# GarmentLab: A Unified Simulation and Benchmark for Garment Manipulation

**Haoran Lu**[1,2*] **Ruihai Wu**[1*] **Yitong Li**[1,3*]

Sijie Li[1,2]    Ziyu Zhu[1,2]    Chuanruo Ning[1]    Yan Shen[1]

Longzan Luo[1,2]    Yuanpei Chen[1]    Hao Dong[1]

[1]CFCS, School of CS, PKU    [2]School of EECS, PKU    [3]Weiyang College, THU

## Abstract

Manipulating garments and fabrics has long been a critical endeavor in the development of home-assistant robots. However, due to complex dynamics and topological structures, garment manipulations pose significant challenges. Recent successes in reinforcement learning and vision-based methods offer promising avenues for learning garment manipulation. Nevertheless, these approaches are severely constrained by current benchmarks, which offer limited diversity of tasks and unrealistic simulation behavior. Therefore, we present **GarmentLab**, a content-rich benchmark and realistic simulation designed for deformable object and garment manipulation. Our benchmark encompasses a diverse range of garment types, robotic systems and manipulators. The abundant tasks in the benchmark further explores of the interactions between garments, deformable objects, rigid bodies, fluids, and human body. Moreover, by incorporating multiple simulation methods such as FEM and PBD, along with our proposed sim-to-real algorithms and real-world benchmark, we aim to significantly narrow the sim-to-real gap. We evaluate state-of-the-art vision methods, reinforcement learning, and imitation learning approaches on these tasks, highlighting the challenges faced by current algorithms, notably their limited generalization capabilities. Our proposed open-source environments and comprehensive analysis show promising boost to future research in garment manipulation by unlocking the full potential of these methods. We guarantee that we will open-source our code as soon as possible. You can watch the videos in supplementary files to learn more about the details of our work. Our project page is available at: `https://garmentlab.github.io/`

## 1 Introduction

The next-generation assistant robots should possess not only the abilities to separately manipulate a wide variety of objects, including rigid, articulated[59], and deformable objects[58], but also the capability to leverage interactions between those physical media, including flow and fluids, in order to assist humans[39]. Among various daily tasks [69, 59, 56], garment manipulation stands out as one of the most challenging, crucial, and extensively discussed tasks in the robotics and computer vision, due to its demanding requirements for understanding dynamic properties of physical instances and interactions between them. For instance, washing clothes entails the interaction between garments and fluids, while dressing up requires collaboration between robots and humans.

Garment manipulation tasks mainly presents three challenges. Firstly, each individual garment possesses nearly infinite states and exhibits complex kinematic and dynamic properties. *Therefore, it is crucial for models to comprehend the various self-deform states of garments, which usually requires*

---

*Equal contribution.

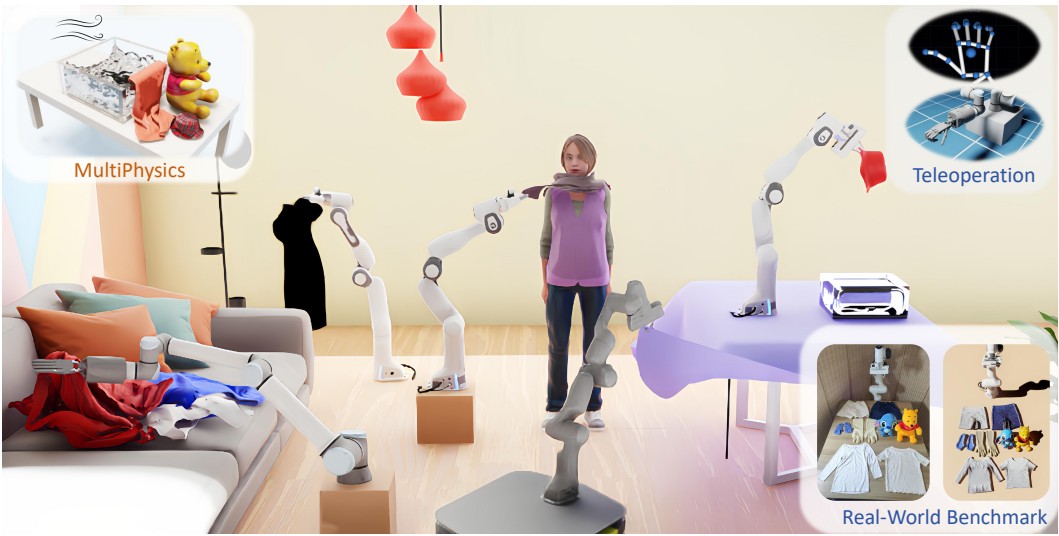

Figure 1: **GarmentLab** provides realistic simulation for diverse garments with different physical propoerties, benchmarking various novel garment manipulation tasks in both simulation and the real world.

*large amount of training data [17, 6]* **(C1)**. Secondly, garment manipulation involves interactions with various types of objects, including rigid (*e.g.*, clothes hanger) and articulated (*e.g.*, wardrobe) objects, as well as fluids and human body. *Consequently, enabling models to understand these interactions across diverse physical media presents great significance* **(C2)**. Finally, considering that strategies for manipulating garments are often highly complex, and visual perception of garments is more challenging due to their diverse states and patterns, *manipulating garments faces a greater sim2real gap [63, 29]* **(C3)**.

Training a powerful agent capable of overcoming these challenges necessitates a vast amount of data encompassing robot-object interactions. However, directly collecting data from the real world is impractical. Thus, researchers have long pursued benchmarks for garment manipulation [30, 4, 6, 64]. Current deformable simulations suffer from drawbacks such as missing garment meshes [30]. Additionally, they offer a limited range of tasks, hindering further research endeavors.

Therefore, we present **GarmentLab** (Figure 1), a unified environment and benchmark for garment manipulation. **GarmentLab** has three novel components to satisfy the demands for diversity and realism: The powerful **GarmentLab Engine**, which is built upon Omniverse Isaac Sim [71] and supports a variety of physical simulation methods. The simulator not only supports Position-Based-Dynamic (PBD) [3], Finite-Element-Method (FEM) [11], to simulate garments, fluid and deformable objects but also makes integration with ROS [42] to provide an efficient teleoperation pipeline for data collection. **GarmentLab Assets** is a large-scale indoor dataset comprising 1) garments models covering 11 categories of daily garments from ClothesNet [70] 2) various kinds of robot end-effector including gripper, suction and dexterous hands. 3) high-quality 3D assets including 20 scenes and 9000+ object models from ShapeNet [7]. Based on realistic simulation and rich assets, we propose **GarmentLab Benchmark** containing 20 tasks divided into 5 groups to evaluate state-of-the-art vision-based and reinforcement learning based algorithm.

To tackle above challenges, our environment has three characteristics:1) **Efficient**. Garment manipulation involves nearly infinite object state and action spaces, requiring substantial data for models to understand garment structure and deformation. To meet this demand, our highly parallelized GPU-based simulator provides a significant training advantage. Larger batch sizes enhance RL-based algorithms [33], while faster data collection speeds reduce training time for perception-based algorithms (tackling **C1**). 2) **Rich**. The richness of our simulator can be categorized into two aspects: the diversity of **simulation content** offered by GarmentLab Assets and the depth of **physical interaction** facilitated by GarmentLab Engine. We emphasize multi-physics simulation, encompassing rigid-articulated, deformable-garment, fluid dynamics, and flow, along with their interactions. This focus is vital for training agents capable of comprehending real-world physical properties [48] (tackling **C2**). You can refer to videos in supplementary material for our simulation effects. 3) **Real**.

Table 1: **Comparisons with Other Deformable Object Environments.**

| | Multi-Camera Scenes | Robot | Sim2real | Garment | Fluid | Rigid Body | Articulated | Human | FEM Objects | Flow | RealBenchmark | Physics | Thermal(stream/fire) | Mobile Task | Dexterous Task | Rendering | Pipeline | Area |
|---|---|---|---|---|---|---|---|---|---|---|---|---|---|---|---|---|---|---|
| Softgym[30] | ✗ | ✗ | ✗ | ✓ | ✓ | ✓ | ✗ | ✗ | ✗ | ✗ | ✗ | ✗ | ✗ | ✗ | ✗ | R | GPU | Manipulation |
| Orbit[34] | ✓ | ✓ | ✓ | ✓ | ✗ | ✗ | ✓ | ✗ | ✗ | ✗ | ✗ | ✗ | ✗ | ✗ | ✗ | RT | GPU | Industrial |
| Sapien[62] | ✓ | ✗ | ✓ | ✗ | ✗ | ✗ | ✓ | ✓ | ✗ | ✗ | ✗ | ✗ | ✗ | ✗ | ✗ | R | CPU | Manipulation |
| Habitat 3.0[39] | ✗ | ✓ | ✓ | ✗ | ✗ | ✗ | ✓ | ✓ | ✗ | ✗ | ✗ | ✗ | ✗ | ✗ | ✗ | R | CPU | Navigation |
| Behavior[28] | ✗ | ✓ | ✓ | ✓ | ✗ | ✗ | ✓ | ✗ | ✗ | ✓ | ✗ | ✗ | ✗ | ✓ | ✗ | RT | GPU | Manipulation |
| Mujoco[53] | ✓ | ✗ | ✓ | ✗ | ✗ | ✗ | ✓ | ✓ | ✗ | ✗ | ✗ | ✗ | ✗ | ✗ | ✗ | R | CPU | Manipulation |
| Pybullet[12] | ✓ | ✗ | ✓ | ✗ | ✗ | ✓ | ✓ | ✓ | ✗ | ✗ | ✗ | ✗ | ✗ | ✗ | ✗ | R | CPU | Manipulation |
| RLBench[23] | ✓ | ✗ | ✓ | ✗ | ✗ | ✗ | ✓ | ✓ | ✗ | ✗ | ✗ | ✗ | ✗ | ✗ | ✗ | R | CPU | Manipulation |
| Gibson[60] | ✓ | ✓ | ✓ | ✗ | ✗ | ✗ | ✓ | ✓ | ✗ | ✗ | ✗ | ✗ | ✗ | ✓ | ✗ | R | CPU | Navigation |
| DeformableRavens[47] | ✓ | ✗ | ✓ | ✗ | ✗ | ✗ | ✓ | ✗ | ✓ | ✗ | ✗ | ✓ | ✗ | ✗ | ✗ | R | CPU | Manipulation |
| **GarmentLab** | ✓ | ✓ | ✓ | ✓ | ✓ | ✓ | ✓ | ✓ | ✓ | ✓ | ✓ | ✓ | ✓ | ✓ | ✓ | **RT** | **GPU** | Manipulation |

As the sim-to-real gap emerges as the main obstacle in developing embodied agents, GarmentLab Engine surpasses Omniverse capabilities by providing mature sim-to-real algorithms, such as Teleoperation [41] utilized in the RL field, and the Visual Sim-Real Alignment Algorithm employed in perception algorithms. We also make integration with ROS [42] and MoveIt [10], which is beneficial for narrowing sim2real gap by introducing real-world robot motions into simulation (tackling **C3**).

Our benchmark experiments highlight the significant challenges current algorithms face, even with seemingly simple tasks like unfolding. These difficulties arise from a lack of understanding of physical interactions and the complexities of high-dimensional states. Vision-based algorithms demonstrate limited generalization, with performance strongly affected by the initial state of objects. RL-based algorithms also encounter difficulties with tasks requiring long-horizon planning. These insights offer valuable guidance for improving methods for garment and deformable object manipulation.

In summary, we have made the following contributions in **GarmentLab**:

- We propose **GarmentLab Environment**, a **realistic and rich environment** for garment manipulation, featuring diverse simulation methods, assets, object physics and multi-material interactions.
- Based on GarmentLab Environment, we propose **GarmentLab Benchmark**, benchmarking a large variety of garment manipulation tasks, and providing the first real-world garment manipulation benchmark that can be reproduced internationally.
- We **integrate different sim2real methods** into GarmentLab, providing solutions to narrowing the sim2real and further facilitating the real-world applications.
- **Extensive experiments and detailed analyses** of different types of garment manipulation algorithms facilitate and enlight future research on garment manipulation.

## 2   Related Work

**Garment and Deformable Object Benchmarks**. Current deformable object environments [30, 61, 63] usually support only one simulation method (*e.g.*, PBD or FEM), limiting the types of simulated objects and interactions. Besides, most environments are CPU-based [62, 63, 12], severely limiting parallel capabilities and often exhibiting a huge sim-to-real gap due to the absence of comprehensive sim-to-real algorithm designs. In contrast, as a GPU-based simulator, GarmentLab provides diverse 3D meshes and supports various simulation techniques. We further integrate ROS and combine it with our carefully designed sim-to-real pipeline, offering a more comprehensive solution for researchers. Detailed comparisons between our environment and others can be found in Table 1 and Appendix C

**Garment and Deformable Object Manipulation**. Although current efforts excel at specific tasks such as folding[63, 1, 57] and unfolding[17, 6], many real-world tasks are long-horizon and involve interactions between various physical media. While many studies have potential to tackle these problems[48, 29], they are hindered by the lack of a mature simulation platform capable of supporting such diverse and complicated extensions. Furthermore, while current research predominantly emphasizes gripper manipulation tasks[2, 65], we introduce tasks involving suction, dexterous hands, and mobile robots. We believe GarmentLab will make a unique and valuable contribution to the robotics community by providing a new platform for developing garment manipulation algorithms and significantly expanding the scope of existing methods.

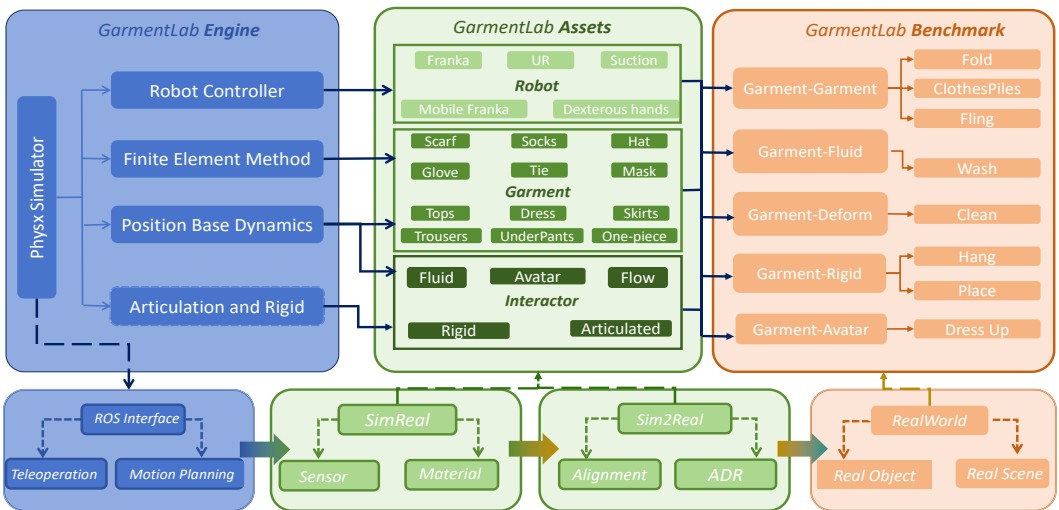

Figure 2: **The Architecture of GarmentLab. (Left)** Built on PhysX5, our environment supports various simulation methods. **(Middle)** Our environment can deliver realistic simulations of diverse robots, garments, and interactions between multiple physics media. **(Right)** Subsequently, we can utilize these assets to construct tasks across various categories. **(Bottom)** The framework supports real-world deployment.

# 3 GarmentLab Environment

GarmentLab aims to integrate state-of-the-art physical simulation methods, modern graphics rendering engines, and user-friendly robotic interfaces into a unified framework (Figure 2). Below we will first introduce *GarmentLab Engine* (Section 3.1) and *GarmentLab Asset* (Section 3.2) to show our diversity in function and objects. As we especially focus on the exploration of multiple physical simulation methods and interaction between them, we will introduce *GarmentLab Physics* in Section 3.3. In section 4 We will talk about our novel-proposed tasks.

## 3.1 GarmentLab Engine

Built on NVIDIA's IsaacSim[71], GarmentLab offers a highly-paralleled data collection pipeline, realistic rendering, support for various sensors, and integration with Robot Operating System (ROS) [42].

**Data Pipeline.** Data pipeline mainly consists of two components: Visual Data System and RL-Training System. Visual System provides both RGB-D observations and ground-truth semantic label including 2D and 3D bounding box, normals and instance segmentation. Based on IsaacGym, RL System can establish multiple agents on GPU at the same time for efficient training.

**Rendering.** GarmentLab supports multiple camera angles, such as eye-on-hand and eye-on-base perspectives, unlike the single-camera setups of past works [30, 61],which employing naive OpenGL framework[49]. Additionally, it utilizes GPU-enabled ray tracing for rendering, which enhances realism and challenge by creating more realistic shadows and lighting [51], thus reducing the sim2real gap and improving the performance of visual algorithms and mobile navigation tasks.

**ROS.** ROS[42] is a generic and widely-used framework for building robot applications. We use ROS to align robot in realworld and the simulation, please refer to Section 6.2 for detailed. Also, although IsaacSim provides traditional Inverse-Kinematic[36] and RMPFlow control[27], we also provide MoveIt framework[10] for motion planning, which is more widely used in the real world.

**Sensor.** In addition to RGB-D observations, auxiliary observations can be accessed, such as robot joints, cloth particles and object poses. They are required in common RL framework and teacher-student network[15]. Other Omniverse sensors (e.g., tactile, contact-report) could also be available.

## 3.2 GarmentLab Assets

GarmentLab Asset compiles simulation content from a variety of state-of-the-art datasets, integrating individual meshes or URDF files into complete, simulation-ready scenes with robots and sensors. We employ Universal Scene Description files to store all assets with attributes, including physics, semantics, and rendering properties. Key components along with their sources and categories are shown in Table 2. More details about each asset type are provided in Appendix A.

Table 2: **Key Components of GarmentLab Assets.**

| Asset Type | Sources | Categories |
|---|---|---|
| Garment and Cloth | ClothesNet | Hats, Ties, Masks, Gloves, Socks, Dishcloths, ... |
| Rigid and Articulated | ShapeNet, PartNet, YCB, PartNet-Mobility | Chairs, Boxes, Washing Machines, Storage Furniture, ... |
| Robot | - | Franka, UR5, RidgebackFranka, ShadowHands |
| Human Model | Omniverse HumanModel | - |
| Materials | Omniverse Base Material | Fabric, Carpet, Leather, Silk, Cotton, ... |

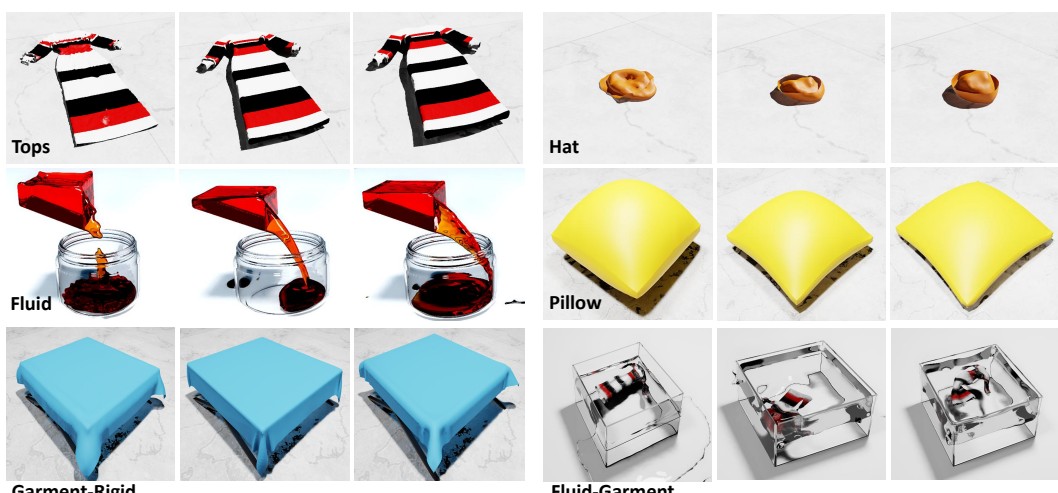

Figure 3: **GarmentLab Physics.** GarmentLab explores the potential of different simulation methods, and provides different physical parameters, modeling the distinct properties of different materials in the real world.

## 3.3 GarmentLab Physics

**Simulation Method.** To ensure physically realistic simulation, we use tailored methods for different objects based on their physical characteristics. For **large garments and fluid**, we use Postion-Based Dynamics (PBD)[3]. For **small elastic garments** like gloves and socks, and everyday objects like toys and sponges, we apply Finite Element Method (FEM)[11]. **Human simulation** involves articulated skeletons with rotational joints and a surface skin mesh for high-fidelity rendering. **Robot simulation** utilizes PhysX articulation system for precise force control, P-D control, and inverse dynamics. Unlike previous works that rely on a single simulation method, GarmentLab provides platforms for exploring dynamics and kinematics of various objects and the coupling and interactions among them. **Diverse Physics Parameters.** To fully exploit the potential of various simulation methods and make garment simulation more diverse and realistic, GarmentLab provides various physics parameter configurations. For example, as cloth is modeled as a grid of particles, altering parameters such as particle size and stiffness will change garment physical behaviors. Likewise, as depicted in Figure 3, diverse physical material parameters are assigned to diverse objects. These parameters encompass, but are not limited to, surface tension and cohesion for fluids, friction for rigid objects, and modulus. It is worth noting that parameters influencing the interaction between different objects, including contact offset and reset offset, are also adjusted. For further details, please consult Appendix D.

## 4 GarmentLab Benchmark

**GarmentLab Benchmark** is motivated by the abilities that an intelligent manipulator agent should possess, including (1) understanding the physics of object interactions, (2) generating accurate action sequences for long-horizon, complex tasks, and (3) transferring this knowledge to the real world. To test these abilities, we classified tasks into five categories based on physical interactions. We also proposed several complex, long-horizon tasks to advance future research. The demos of tasks and real-world experiment are shown in figure 4.

**Task Categories.** To fully exploit the model's capability in understanding physical interactions and conduct comprehensive evaluations of current algorithms, we categorize 20 tasks into 5 groups. The example tasks and corresponding categories are shown in Table 3. Examples of task sequences are provided in Figure 4. You can refer to Appendix G to get more details.

**Long-Horizon Tasks.** With the advancement of robotics, there is a growing focus on completing long-horizon tasks, which integrate skills tasks such as 3D perception, manipulation and navigation. Thus we propose several long-horizon garment manipulation tasks, including *organizing clothes*, *wash clothes*, *make up tables* and *dress up*. These tasks go beyond simply executing subtasks in sequence, as they require holistically planning how to accomplish the task based on the environment. During the execution, the algorithm needs to consider the positioning of the operation, the placement location, and carefully plan the path to avoid collisions. More analysis are shown in Section 7.

Table 3: **Task Categories of GarmentLab Benchmark.**

| Task Type | Focus | Example |
|---|---|---|
| Garment-Garment | fundamental garment manipulation | fold, unfold, clothes piles retrieval... |
| Garment-Fluid | interaction between garments and fluids as well as flow, | Washing clothes, Drying clothes with a hairdryer... |
| Garment-FEMObjects | manipulating FEM Objects | Packing clothes, Dexterous grasp plush toy |
| Garment-Rigid | common interactions between clothing and rigid bodies | Hanging, Putting clothes into washing machine |
| Garment-Avatar | collaboration with human | Putting a scarf, Dress a person in T-shirt |

## 5 Real-World Benchmark

Real-world benchmark is crucial for not only evaluating the real-world performance of different methods, but also providing a standardized platform for researchers to reproduce and exchange methods. With the existence of real-world dataset or benchmarks for rigid [5], articulated [31] objects and furnitures [18], we introduce the first real-world benchmark for deformable objects and garments.

Unlike rigid or articulated objects that can be 3D-printed from CAD files, deformable objects are usually purchased without CAD files. Easily influenced by external forces, it is difficult to accurately model garments directly using traditional multi-camera calibration and surface reconstruction methods. Therefore, we use commercial scanning devices with lasers and light for mesh scanning.

Selected objects cover diverse garments (tops, trousers, socks, hats), plush toys, household items (bags, clutches), and cleaning supplies. They are primarily selected from well-known international brands for durability and accessibility. To ensure variety, objects have different shapes, sizes, transparencies, deformabilities, and textures. For instance, our dataset features various tops made from materials like assault jackets, down jackets, shirts, and vests, with a wide range of physical attributes.

Additionally, we provide semantic human annotations for object part masks and key points, supporting dexterous manipulation such as grasping specific parts and object tracking using key points. Following YCB[5], we present a systematic approach for defining manipulation protocols and benchmarks. These protocols specify the experimental setup for each task and provide procedural guidelines. A comprehensive description of the real-world benchmark is provided in Appendix F.

## 6 Sim2Real Framework

Transferring models from simulation to reality is crucial yet challenging. GarmentLab paves way to realistic application by integrating methods for mitigating vision (Sec. 6.1) and action (Sec. 6.2) gap.

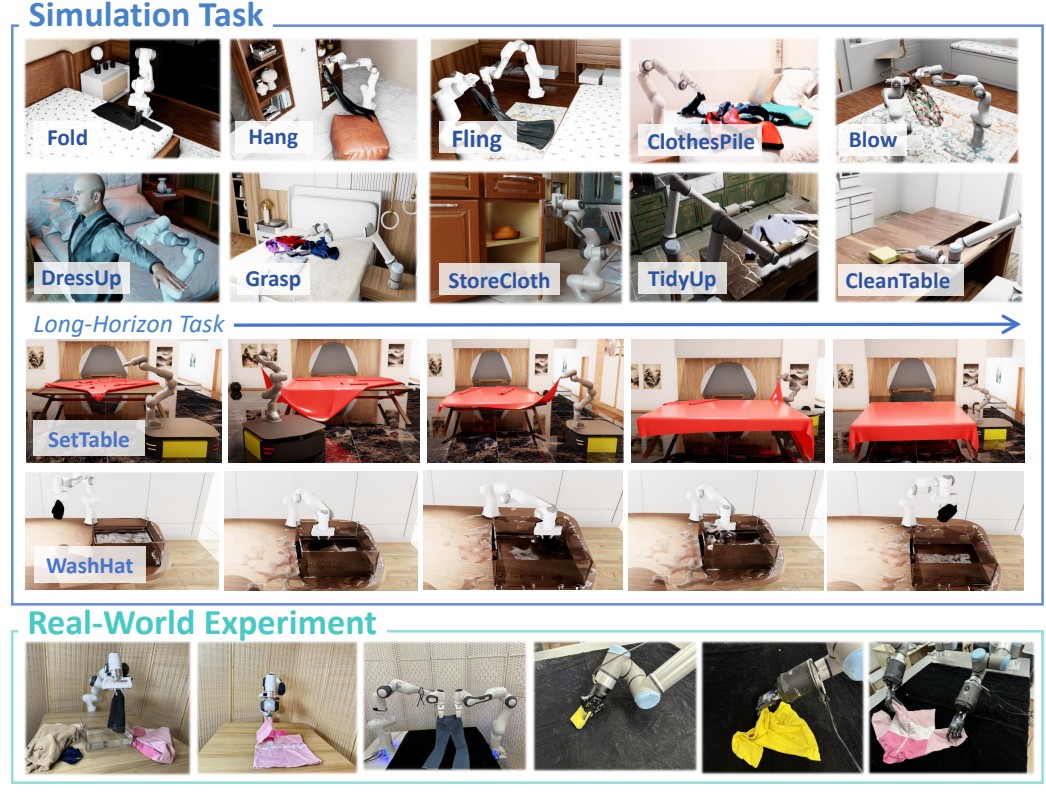

Figure 4: **Diverse Tasks of GarmentLab Benchmark.** We introduced 20 garment and deformable manipulation tasks including complicated long-horizon tasks. The last row shows the execution of these tasks in the real world.

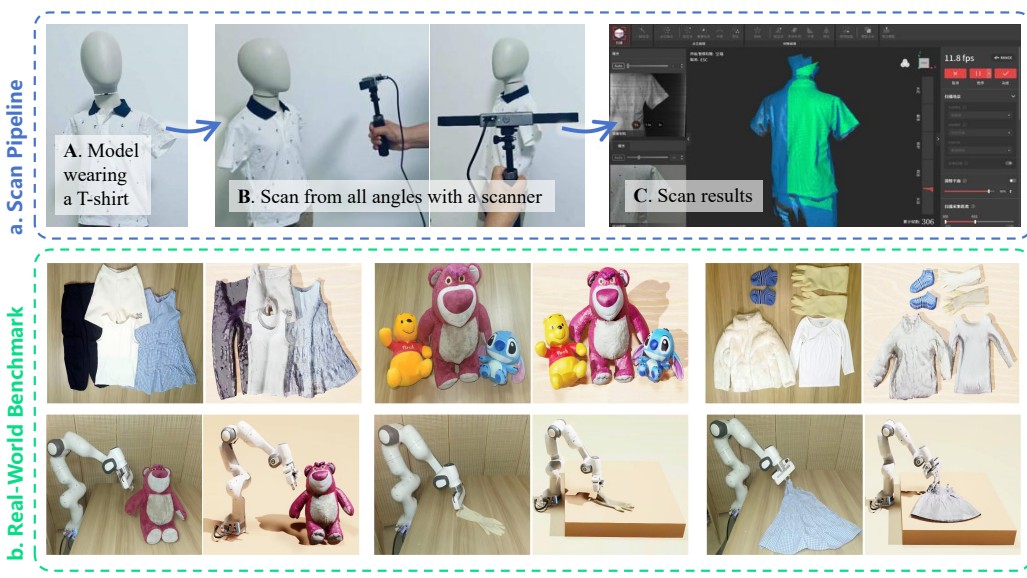

Figure 5: **Real-World Benchmark.** Part a demonstrates the whole pipeline of converting real-world objects into simulation assets. Part b demonstrates the performance of different categories of objects in both simulation and the real world (the first row), and the results of these objects being manipulated by the robot (the second row).

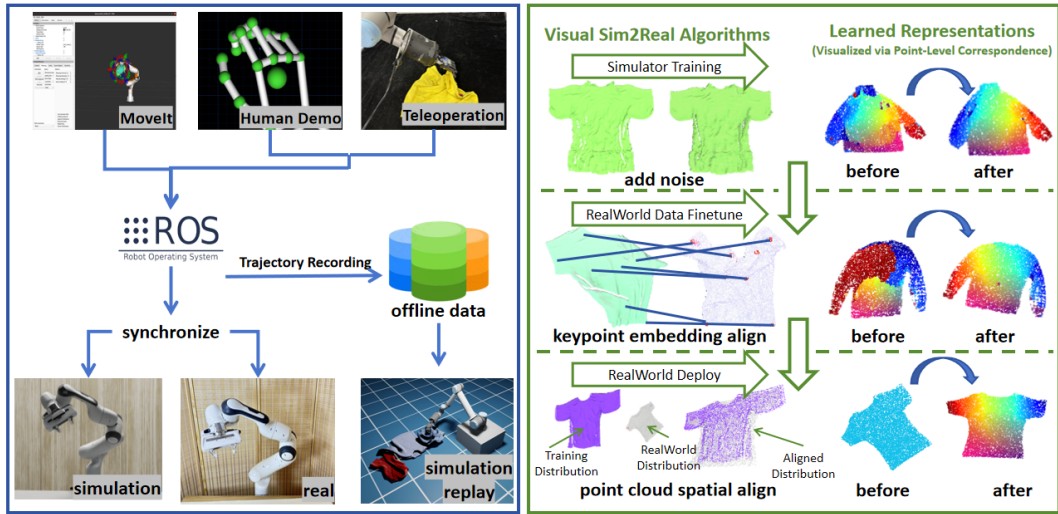

Figure 6: **Sim2Real Framework.** On the left, we highlight our MoveIt and teleoperation pipeline, a lightweight and easy-to-deploy system built using ROS. On the right, we present our three proposed visual sim-to-real algorithms, demonstrating a significant improvement in model performance after deploying these algorithms.

## 6.1 Sim-Real Vision Alignment

GarmentLab intergrates several automated and self-supervised sim2real methods, and have verified their effectiveness by predicting dense visual correspondence for manipulation [57] (Figure 6, Right), with quantitative manipulation success rate in Table 6.

**Keypoint Embedding Alignment.** Aligning corresponding skeleton point representations can mitigate representation gap between point cloud in simulation and the real world [57]. By attaching markers to skeleton points and enabling robot to perform self-play, we obtain ground-truth keypoint pairs and employ InfoNCE [25]to align corresponding point representations. Shown in Figure 6, the alignment adapts representations to the real-world distribution. Appendix E shows more details.

**Noisy Observation.** Adding noise to point cloud for training can be very effective for sim2real transfer [16]. As shown in Figure 6, initial query results had many errors. By adding noise during training, our model became more robust, leading to smoother and more accurate representations.

**Point Cloud Alignment.** We propose aligning point clouds by optimizing an affine matrix, using chamfer distance as the loss function. As shown in Figure 6, the model initially predicts incorrect results, even for flat surfaces. However, after alignment, it successfully predicts accurate results.

## 6.2 Real-World Motion Generation

For many algorithms, action trajectories generated in simulation is not align with those in the real world. We introduce two methods for generating trajectories in simulation that closely mimic real-world scenarios by leveraging prior knowledge of real-world manipulation trajectories.

**Teleoperation.** We've developed a lightweight, cost-effective teleoperation system requiring just one-click deployment. It facilitates simultaneous control of dexterous hands and grippers in both real-world and simulated settings (Figure 6). This system supports data collection for offline training, like diffusion policy.[67, 9]. Implementation details are in Appendix I

**MoveIt.** Incorporating MoveIt into our framework elevates motion planning and obstacle avoidance beyond heuristic trajectory methods, as noted in previous studies [57, 63]. Employing MoveIt for real-world robot execution also aids visual algorithms. Adapting models to MoveIt-generated trajectories during training reduces the sim-to-real gap. Detailed implementations are provided in Appendix H.

# 7 Experiments

## 7.1 Simulation Experiment Setup

**Methods.** We selected three vision-based and two reinforcement learning (RL) algorithms for experiment, with details listed in Table 4. For vision-based algorithms, we prioritized those utilizing dense representations for garments, as they have demonstrated generalization ability and are suitable for various downstream tasks.

Table 4: **Benchmark Methods of GarmentLab.**

| Method | UniGarmentManip (UGM)[57] | DIFT[52] | Affordance[58] | RL-State[46] | RL-Vision[46] |
|---|---|---|---|---|---|
| Type | 3D-Visual-Correspondence | 2D-Visual-Correspondence | 3D-Representation | RL | RL |
| BackBone | PointNet++[40] | Stable-Diffusion | PointNet++[40] | PPO | PPO |
| Input | Point Cloud | RGB Image | Point Cloud | State-Base GT | Partial Point Cloud |

**Tasks.** Although we proposed many novel tasks, current algorithms cannot fully solve them. Thus, for large garments like tops, dresses, and trousers, we chose folding, hanging, and unfolding tasks. For small items like hats and gloves, we selected hanging and placing tasks to evaluate visual and RL algorithms. For dexterous and mobile tasks, existing work mostly employs RL algorithms. Hence, we evaluated the performance of both RL-state-based and RL-vision-based algorithms separately.

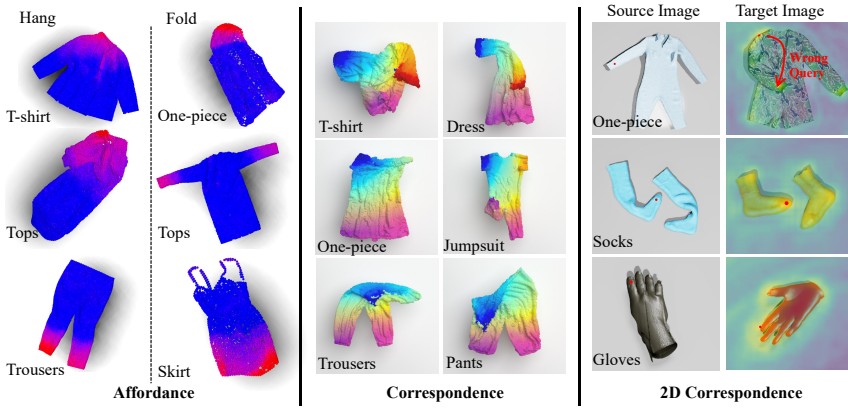

Figure 7: **Qualitative Results.** We visualize the qualitative results of the three vision-based algorithms: the left, middle, and right sections of this image correspond to the Affordance, Correspondence, and DIFT algorithms, respectively. Note that the DIFT exhibits query errors. For detailed analysis, please refer to Experiment Section.

## 7.2 Simulation Result and Analysis

Table 5 present quantitative comparisons between vision-based algorithms and RL-based algorithms. Figure 7 intuitively demonstrates the visual results of three different vision algorithms.

Table 5: **Simulation Results on Traditional Tasks.** Numbers in the Large-piece column represent scores for Tops, Trousers and Skirts. Numbers in the Small-piece column represent scores for Hats and Gloves.

|  | Large-piece | | | Small-piece | |
|---|---|---|---|---|---|
| Method | Fold | Unfold | Hang | Place | Hang |
| UGM | **61.5 / 62.1 / 59.8** | **58.3 / 60.5 / 57.2** | 61.8 / 57.5 / 59.7 | 33.2 / 35.4 | 31.8 / 29.2 |
| DIFT | 32.7 / 36.7 / 31.2 | 18.7 / 23.3 / 17.6 | 31.2 / 27.6 / 29.7 | **66.4 / 63.2** | **64.7 / 61.2** |
| Affordance | 53.2 / 51.8 / 56.9 | 32.4 / 36.7 / 31.8 | **64.1 / 60.2 / 61.3** | 63.2 / 61.5 | 62.6 / 60.4 |
| RL-State | 14.8 / 12.5 / 9.8 | 6.5 / 8.8 / 12.7 | 13.1 / 19.7 / 14.7 | 14.2 / 12.1 | 12.8 / 13.2 |
| RL-Vision | 6.7 / 8.2 / 3.2 | 5.2/ 6.2/ 8.8 | 7.6 / 5.3 / 4.1 | 13.1 / 14.8 | 11.3 / 15.2 |

**Vision-Based Algorithm.** Among the three vision-based algorithms, UGM performed best on large-piece clothing, emphasizing cross-deform and cross-object consistency in learning representations, DIFT excels with small-piece clothing due to its robustness to object rotation but lacks proficiency in understanding clothing folding. Affordance works well for tasks that do not require precise point selection, such as hanging, but struggles with folded garments.

**RL Algorithm.** Compared to vision-based algorithms, RL performs poorly on garment manipulation due to the complex dynamics of garments. Our analysis of training videos showed that RL often generates abnormal trajectories, causing clothes to get tangled with the robotic arm or be pushed away. This issue is more pronounced with RL-vision-based methods, as the higher-dimensional and partial visual observations hinder the model's ability to converge on an effective strategy. For dexterous and mobile tasks, the larger action and search spaces result in suboptimal performance. Further discussion and analysis can be found in Appendix B.

### 7.3 Real-World Experiments

In our real-world experiments, we focused on testing vision-based algorithms due to the risk associated with RL actions. T-shirts for folding and hats for hanging, were selected for experimentation. Additionally, we conducted ablation study on proposed sim2real methods using UGM (Table 6). Our real-world results align with our simulation findings, indicating GarmentLab environment can enhance real-world applications. For sim2 real algorithm, without point cloud alignment and noise augmentation along with keypoint embedding alignment can improve representation smoothness and accuracy. Qualitative sim2real results are shown in Figure 6 (Right).

Table 6: **Real-World Experiment and Ablation Results**, w/o PA, w/o Noise, and w/o EA respectively represent UniGarmentManip without Pointcloud Alignment, Noise Augmentation, and KeyPoint Embedding Alignment.

| Method | UGM | DIFT | Affordance | w/o PA | w/o Noise | w/o EA |
|---|---|---|---|---|---|---|
| Tshirt-Folding | **10**/15 | 8/15 | 6/15 | 2/15 | 8/15 | 7/15 |
| Hat-Hanging | 10/15 | **14**/15 | 9/15 | 5/15 | 8/15 | 9/15 |

## 8 Conclusion

We introduce GarmentLab, a comprehensive environment and benchmark for manipulating garments and deformable objects. GarmentLab includes the GarmentLab Engine, supporting various simulation methods and ROS integration; GarmentLab Assets, a diverse dataset of robots, materials, and garments; and GarmentLab Benchmark, proposing several novel tasks. It also provides the first real-world deformable benchmark along with several sim2real methods.

## 9 Acknowledgment

This project is supported by The National Natural Science Foundation of China (No. 62376006), the National Youth Talent Support Program (8200800081) and the National Natural Science Foundation of China (No. 62136001).

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

# Appendix Overview

## A GarmentLab Asset

GarmentLab asset contains a vast range of objects which are commonly seen in our daily life, from rigid object, articulated object, garment with various materials to all kinds of robots and human models.

## B Experiment

Details about how the experiments described in the main paper is conducted, such as partition of training and testing sets and evaluation metrics definition.

## C Related Work

The main reference work under the topic of traditional embodied and robotic simulator, deformable objects and cloth Benchmark, downstream tasks and algorithms of garment manipulation.

## D Physics Simulation

Details about the simulation methodology of different types of physics simulation, from garment, deformable body, rigid body to wind and fluid. Discussion of how the changing of physics parameters leads to different simulation performance.

## E Sim2Real

More detailed description of sim-to-real visual alignment and motion generation methods with specific algorithm process and quantitative mathematical formulas.

## F Real-World Benchmark

Discussion about the principle and consideration of object selection. Detailed process of scanning real-world objects into simulation models and protocol benchmark guidelines of garment manipulation.

## G Task

The task categories and detailed settings with multi-physics interactions, including but not limited to garment-garment task, garment-avatar task. Long-horizon tasks are also discussed here.

## H MoveIt

How trajectories generated by MoveIt are recorded and how to adapt visual models in the simulator to the trajectories, aiming to narrow sim-to-real gap.

## I Teleoperation

Detailed process of how the motion of human demonstration is captured and then adapted to embodiment, bringing robots with action in human-level intelligence.

## J Limitation

Limitation of our work.

## K Broader Impact

The potential societal impacts of our work.

# A  GarmentLab Assets

- **Rigid Object** We mainly import objects from ShapeNet[7],PartNet[35]and YCB dataset[72]. Note that we have filtered out objects that are not suitable for physical simulation and have issues interacting with garments or fluid, and then reorganized and reclassified the dataset.

- **Articulated Object** Having much higher degree-of-freedom(DoF) state spaces, articulated objects are, however, generally more difficult to understand and interact with compared to 3d rigid objects. We mainly import articulated objects from PartNet-Mobility dataset [62] including Chair, Box, Bucket, Washing machine and Storage Furniture etc, to establish the comprehensive tasks for indoor robots such as folding clothes and putting them into the wardrobe.

- **Garment and Cloth** We select garments from ClothesNet [70], a large-scale dataset of 3D clothes objects with information-rich annotations. We select garments from 11 categories including Hat, Tie, Mask, Gloves and Socks and use two physical simulation method to simulate them. We also include standard square cloths, like dishcloths and tablecloths, to cover indoor task needs. Note that there are still gaps between meshes and ready-to-simulate object, we do post-processing of garments including giving correct physical parameters to simulate them.

- **Robot**. We deploy a variety of specialized robots for diverse tasks, including a 7-DoF Franka manipulator[37] with a parallel gripper, a UR5 with suction for manipulation tasks, and a RidgebackFranka[45] with wheels for mobility and navigation. For dexterous tasks, we use ShadowHands[44] mounted on a UR10e.

- **Human Model**. We incorporate human model to construct long-horizon tasks, such as dressing up. Utilizing avatars selected from actorcore[43], we assign specific motions to each avatar to facilitate collaboration with robots. Each avatar comprises articulated joints, surface skin mesh, and clothing, enabling realistic simulation of human structure and motion.

- **Materials**. Materials are crucial components of virtual relightable assets, defining the interaction of light at the surface of geometries. We carefully choose materials from the Omniverse Base Material library to attain optimal rendering outcomes, a critical aspect for visual-based algorithms. Moreover, diverse textures can aid algorithms in understanding the relationship between an object's appearance and its physical behavior.

# B  Experiment

## B.1  Overview

**Generalization ability.**  As a novel environment, **GarmentLab** especially focus on evaluating and improving the generalization abilities of algorithm. We evaluate the generalization ability from the following aspects. **Novel Object** Thanks to rich GarmentLab Asset, we split garment and other object dataset into *Train/Var/Test* at proportion 70%/15%/15% to test algorithms generalization ability on object level. Moreover, as garment and deformable object have nearly infinite self-deform state, we introduce **Novel State**. For example, we disturb garment initial state and test model's ability on handling wrinkled and folded clothes. Moreover, in order to improve algorithm ability of planning and collision avoidance, we also involve **Novel Scene**, as for task like make up table, the shadow of irrelevant object can also influence navigation.

**Metrics.**  We primarily use the success rate as the evaluation metric. It is important to note that because garments and fluids can easily change state due to gravity or friction, we consider a task successful if it meets the success criteria and maintains this state for at least five seconds. For tasks that appeared in previous work, we adopted the widely accepted success criteria and tolerance thresholds from the goal state. For example, we use Intersection-over-Union (IOU) between the target and the folded garments to evaluate folding task[57, 6] and use coverage area to evaluate unfolding task[17]. For novel tasks such as washing or blowing, the goal states and tolerances are derived according to human behaviors.

## B.2 Experiment Task Setting

For large garments like tops, dresses, and trousers, we chose folding, hanging, and unfolding tasks. And for small items like hats and gloves, we selected hanging and placing tasks to evaluate visual and RL algorithms. The detailed experiment settings for the tasks listed above are shown below:

**Garment-hanging** task requires robots to hang garments to a fixed couple. The first criterion of success is that the garment can be hanged steadily (five seconds in real experiment) on the couple. Then to ensure that the garment is hanged in the right pose (not hanged at sleeve or other strange cases), we compare the manipulation result with a standard human demonstration by computing the sum particle-to-particle distance between two states. The total distance within a predefined value will be regarded as success. The initial states of garment to hang is obtained by dropping from random initial poses over the ground.

**Unfolding** task requires robots to unfold garments at random deformations to be flat. Follow ClothFunnel[6] unfolding task success when the garment ground-truth vertices are within a reasonable range of the initial state, i.e., the flat state before the vertices were disrupted. This is because we found that using coverage area as defined by FlingBot[17] may not reasonably reflect success as the garment structure becomes more complex. The initial states of garment to unfold is obtained by dropping from random initial poses over the ground.

**Folding** task requires robots to fold garments from a flat states. After manipulation, we calculate the particle-to-particle distance between a.the final state of garment after manipulation trial and b.the garment state obtained by human demonstration. The manipulation whose total distance is lower than a predefined value can be regarded as success. The initial states of garments to fold are obtained by placing flatly on random position with small disturbance.

**Hat-Hanging** task requires robots to hang hats to a fixed couple. The criterion of success is that the hat can be hanged steadily on the couple and would not fall on the ground. The initial states of hat to hang is obtained by dropping from random initial poses over the ground.

**Placing** task requires robots to get the hats/gloves which are previous hanged at the couple. The placing task succeeds when the hats/gloves are fetched and placed on the right position without falling on the ground. The initial states of hats/gloves to fetch and place are obtained by random dropping from a random poses over the couple, where only the successfully hanged cases will be used for training and testing.

## B.3 Detailed Analysis

**Vision-Based Algorithm.** Comparing the three vision models, we found that UniGarmentManip (UGM) have the best performance on Large-piece of garments. We conjecture that this stems from (1) The consistence of representation on self-deformations. As model have the understanding of deformation on garments, it is easier to detect keypoints required in Folding tasks and Fling tasks in diverse garment states. (2) The explicit design of cross-object representation consistency. UniGarmentManip use skeleton(a graph of keypoints) as the shared bridge for different garments with similar structures, which makes model have the ability to understand the topology of the 3D object in the same category. However, we found that this result is not as good as the original paper using PyFlex. This could be because we introduced the robot and the scene here, so some invalid selection points, such as those beyond the robot's reach or causing collisions, were considered unsuccessful. Additionally, while the original paper used only T-shirts, our study included jackets and other garments with front openings, this wider variety of clothes also increased the complexity of our task. For Affordance, we found that it performs well in Hanging tasks possibly because of its task-specific designed which makes it chooses the grasp points more accurately. In contrast, DIFT has poor performance on these three tasks especially on unfolding tasks due to the unawareness of garment deformations on 2D pretrained correspondence. This is reasonable because most objects in world for training do not have garment-level deformations. However, DIFT perfors better with small-piece garments like hats and gloves due to their minimal deformation, Besides, the pretrain model based on large diffusion model are more robust to rotation, which is crucial for handling small clothes. For UniGarmentManip and Affordance, they are not 3D-equivariant models, so they are more sensitive to rotations, resulting in poorer performance with small clothes compared to DIFT.

**RL-Based Algorithm.** We modified the traditional PPO by directly replacing the value net of PPO with the success information from the simulation ground-truth information. This is because in robotics tasks, the value of the policy can be easily obtained from the ground-truth information.At the same time, following ClothFunnel[6] and FlingBot[17], we primarily used RL for selecting points and adopted a scripted policy for the trajectory, with simple adaptations based on the selected points. This is because directly using RL to train the trajectory is particularly unstable. We will elaborate on this point in more detail below.

We found that PPO's performance on state-based tasks was significantly worse than visual algorithms. After analyzing the training videos, we identified several reasons for this: (1) Abnormal trajectories of the robotic arm caused collisions and pushed the clothes far away. (2) Large reward fluctuations in long-horizon tasks led to training instability, as the robotic arm's random folding actions in the early stages caused significant reward variations. (3) Completing long-horizon tasks was difficult due to the robotic arm's abnormal trajectories disrupting previous steps, such as interfering with the sleeves during the folding task. (4) Wide-ranging movements of the robotic arm caused clothes to wrap around it, particularly in the hanging task, leading to failure. The visual algorithm avoided these issues because its execution trajectories were mostly predefined, such as pick-and-place or fling trajectories.

We also found that the performance of visual-based PPO is significantly inferior to state-based PPO, due to the following reasons: (1) The higher dimensionality of visual input makes training more difficult. (2) The visual input consists of partial point clouds, which can confuse the model, especially for thin objects like clothing. (3) Detecting the object position is more challenging for visual input, resulting in algorithmic failures during the grasping stage. These findings are consistent with those of SoftGym [30].

### B.4 Training Details of Main Algorithms

#### B.4.1 UniGarmentManipulation (UGM)

For **Hyper-parameters selection**, we set batch size to be 32. In each batch, we sample 32 garment pairs. For each garment pair, we sample 20 positive and 150 negative point pairs for each positive point pair. Therefore, in each batch, $32 \times 32 \times 20$ data will be used to update the model. During the Correspondence training stage, we train the model for 40,000 batches. During Coarse-to-fine Refinement, we train the model for 100 batches. During Few-shot Adaptation, we slightly refine the model using 5 demonstration data. Besides, we set the number of skeleton pairs to be 50.

For **computational resource**, we use PyTorch as our Deep Learning framework. Each experiment is conducted on an RTX 3090 GPU, and consumes about 22 GB GPU Memory for training. It takes about 12 hours to train the Coarse Stage, with 1-2 hours of Coarseto-fine Refinement and 0.5 hour's Few-shot Adaptation.

#### B.4.2 Affordance

For **Hyper-parameters selection**, we set batch size to be 128, where each pair contains one positive manipulation point and one negative manipulation point on the same garment, automatically balancing the training data. "Positive" means manipulating on that point can lead to the success of the whole task while "negative" means failure. In each batch, $128 \times 2$ data will be used to update the model. During the Affordance training stage, we train the model for 36,000 batches. The model is designed to make binary classification with cross-entropy as loss function. The output of affordance model reflects the success rate when manipulating on that point, which ranges from 0 to 1. During manipulation, we just select the point with the highest score to manipulate.

For **computational resource**, we use PyTorch as our Deep Learning framework. Each experiment is conducted on an RTX 3090 GPU, and consumes about 16 GB GPU Memory for training. It takes about 18 hours to train the model for a task with a specific category of garment.

#### B.4.3 DIFT

As pretrained model, DIFT use stable-diffusion as backbone. For **Hyper-parameters selection and prompt enginering** We use the default parameters of DIFT. We crop the image size to $762 \times 762$ and set timestep for diffusion to 261. The ensemble size was set to 8. We use official network architecture

pipeline followed by our own designed robot excution pipeline. The robot execution pipeline is similar to UniGarmentManip. We only substitute the query model to DIFT. For **computational resource**, we use PyTorch as our Deep Learning framework. Each experiment is conducted on an RTX 3090 GPU, and consumes about 20 GB GPU Memory for inferencing.

## C  Related Work

**Traditional Embodied and Robotic Simulator** The simulator plays an indispensable role in robotics development as it allows for the rapid and safe acquisition of vast amounts of interaction data, facilitating the implementation of various algorithms. However, the majority of mainstream robot simulators[53, 12, 33, 62] primarily support rigid object simulation including the collision and friction between them. Besides, most of robot simulators are CPU-based[39, 66, 64], severely limiting their parallel capabilities and resulting in slow training speeds. Additionally, these simulations exhibit a significant sim2real gap due to the absence of comprehensive sim2real algorithm designs[20, 38, 19].Nevertheless, based on Isaac Sim[71], our benchmark not only supports parallel data collection but also incorporates comprehensive sim2real designs, including RL-based and vision-based algorithms. **Deformable and Cloth Benchmark** In recent years, there has been a surge in deformable and garment simulation environments[30, 22, 61]. However, the most server problem of these kinds of simulation or benchmarks is that they can only simulate certain kinds of objects as they only support one simulation method, which makes it impossible to explore the physical interaction between multiple kinds of objects. Moreover, these benchmarks are lack of diversity as they are built directly on the underlying simulation architecture and have not integrated with mature platforms, thereby limiting the range of simulated objects and scenes. For instance, softgym[30], built on NVIDIA Flex[32], is confined to simulating tops and trousers while fluidlab[61], built on Taichi[21], can only simulate fluid and performs poorly on rigid objects simulation. Additionally, many benchmarks[61, 30] lack the ability to import robots and establish real grasps, posing significant challenges for joint control and vision-based algorithms. By contrast, GarmentLab provides sophisticated 3D meshes and facilitates various simulation techniques, enabling the modeling of garments, fluids, flow dynamics, avatars, rigid and articulated objects, and their interactions. This inclusive and adaptable platform offers a more comprehensive solution for research and development. The full detailed comparison of out benchmark between others can be found in Appendix A. **Garment and cloth manipulation** Manipulating a single garment or cloth is a well-studied area, with previous works focusing on learning policies for specific tasks such as folding [1, 63], unfolding [17], grasping [8, 68], and dressing-up [55]. However, as many daily tasks involve interactions between various physical media, current algorithms often fall short in solving real-life tasks. Although many proposed algorithms have full potential to solve these problems[59, 29], they are hindered by the lack of a mature simulation platform capable of supporting such simulations. Furthermore, while current research predominantly emphasizes gripper manipulation tasks, we introduce tasks utilizing suction, dexterous hands, and mobile robots. We believe that GarmentLab will make a unique and valuable contribution to the robotics community by providing a new platform for developing garment manipulation algorithms and significantly expanding the scope of existing methods.

## D  Physics Simulation

### D.1  Modeling methodology

#### D.1.1  Position-Based Dynamics (PBD) for Garment

*Position-Based Dynamics (PBD)* is an efficient and stable method for simulating cloth, particularly suitable for complex garments like dresses. PBD models deformable objects as systems of interconnected particles governed by constraints that dictate their physical interactions and behaviors. In PBD, a dress is represented as a triangular mesh where particles serve as discrete points on the cloth surface with attributes such as position $x_i$, velocity $v_i$, and inverse mass $w_i$. The method operates by directly manipulating particle positions to satisfy a series of constraints, achieving stable simulations of deformable materials. These constraints include stretching constraints, which enforce distance maintenance between neighboring particles to prevent excessive elongation, mathematically defined as $C(x_i, x_j) = \|x_i - x_j\| - d$, where $d$ is the rest distance between particles $x_i$ and $x_j$. Bending constraints maintain angles between adjacent triangles in the mesh to simulate resistance to

bending, formulated as $C(x_i, x_j, x_k)$, where the constraint function depends on the angle between particle triplets. Collision constraints detect and resolve collisions between particles and other objects, ensuring realistic interactions within the environment. The PBD algorithm involves initializing particles and constraints based on the dress's geometry and material properties, applying external forces such as gravity, predicting new particle positions as $\hat{p}_i = p_i + \Delta t \cdot v_i$ where $\Delta t$ is the time step, iteratively adjusting particle positions to satisfy constraints, updating particle velocities, and determining final positions for each time step.

### D.1.2 Position-Based Dynamics (PBD) for Fluid Simulation

*Position-Based Dynamics (PBD)* is a powerful method for simulating fluids due to its computational efficiency and stability. PBD treats fluids as collections of particles, where each particle represents a small volume of the fluid. Constraints are applied to ensure physical properties such as incompressibility and realistic fluid behavior. In fluid simulation, particles are characterized by attributes such as position $x_i$, velocity $v_i$, and inverse mass $w_i$.

A key constraint type in fluid simulation is the density constraint, which ensures that the fluid maintains a constant density. The density constraint for a particle $i$ can be defined as:

$$C_i(\mathbf{x}) = \left( \sum_j m_j W(\|x_i - x_j\|, h) \right) - \rho_0$$

where $W$ is the smoothing kernel function, $h$ is the smoothing length, $m_j$ is the mass of particle $j$, and $\rho_0$ is the rest density of the fluid. Collision constraints handle interactions between fluid particles and solid boundaries, ensuring particles do not penetrate solid objects.

The PBD algorithm steps for fluid simulation include initializing fluid particles with positions, velocities, and masses, applying external forces such as gravity, computing predicted positions $\hat{p}_i = p_i + \Delta t \cdot v_i$, adjusting particle positions to satisfy density and collision constraints, updating particle velocities based on the corrected positions, and integrating the updated positions and velocities for the current time step.

### D.1.3 Finite Element Method (FEM) for Simulating Deformable Objects

*The Finite Element Method (FEM)* is a robust numerical technique for simulating the mechanical behavior of deformable objects, ideal for intricate geometries and diverse material properties, such as a toy bear. FEM discretizes the object into a mesh of finite elements and solves the equations of motion to accurately capture realistic deformations under various forces.

In FEM, the deformable object is represented by a mesh consisting of nodes and elements. Nodes are points where the equations of motion are solved, and elements are polyhedral shapes, such as tetrahedrons, that connect these nodes. Material properties, including elasticity, density, and damping, determine the response of the object to applied forces.Modeling a deformable body involves several key steps. First, a deformable body component is added to the mesh, which generates collision and simulation tetrahedral (tet) meshes from the source mesh. The mesh is then separated into visualization, collision, and simulation tetmeshes, each serving distinct purposes in rendering, collision resolution, and simulation. Configuring the material properties involves defining characteristics such as stiffness and dynamic friction by creating and binding a new deformable body material.

### D.1.4 Flow Models for Simulating Wind Effects

*Flow models* are essential for simulating wind effects, capturing the interactions between fluid (air) and objects. These models represent phenomena such as airflow, turbulence, and aerodynamic forces. The typotypoequations form the core of flow models and include the continuity equation for mass conservation $\frac{\partial \rho}{\partial t} + \nabla \cdot (\rho \mathbf{u}) = 0$, where $\rho$ is the fluid density and $\mathbf{u}$ is the velocity vector; the momentum equation for force balance $\frac{\partial (\rho \mathbf{u})}{\partial t} + \nabla \cdot (\rho \mathbf{uu}) = -\nabla p + \nabla \cdot \tau + \rho \mathbf{g}$, where $p$ is the pressure, $\tau$ is the stress tensor, and $\mathbf{g}$ is the gravitational acceleration; and the energy equation for thermal effects $\frac{\partial (\rho E)}{\partial t} + \nabla \cdot (\rho E \mathbf{u}) = -\nabla \cdot \mathbf{q} + \tau : \nabla \mathbf{u} + \rho(\mathbf{g} \cdot \mathbf{u})$, where $E$ is the total energy per unit mass and $\mathbf{q}$ is the heat flux vector.

Wind effects are modeled by defining wind sources, preparing high-resolution meshes, setting boundary conditions, configuring the appropriate flow model (e.g., LES or RANS), and running the simulation to compute wind interactions iteratively. This approach ensures accurate and dynamic representations of wind effects in various environments.

### D.1.5 Rigid Body Simulation

Rigid body models are essential for simulating solid objects that move and interact based on physical laws without deforming. These simulations accurately represent the dynamics of solid objects under various forces. Key components include a rigid body component, which provides properties like linear and angular velocity, and a collision component, which defines how the body collides with other objects.The dynamics of rigid bodies are governed by solvers such as Temporal Gauss-Seidel (TGS) and Projected Gauss-Seidel (PGS), which ensure stability and efficiency. TGS improves convergence by considering temporal aspects of the simulation, while PGS iteratively projects velocities to satisfy constraints.Rigid bodies interact through collisions defined by collision shapes, which can be approximated using convex hulls, bounding shapes, or signed distance fields (SDFs). These approximations balance accuracy and computational performance.Mass properties of rigid bodies are derived from the volume and density of their collision geometries. For more precise control, explicit mass or density values can be set using a Mass component. This allows for accurate simulation of complex interactions and dynamic behaviors.

### D.2 Multi-Physics Simulation Parameters Table

To maximize the value of different simulation methods, we assigned different parameters to various objects. In Table 7, we list all the adjustable parameters.

### D.3 Parameter Effects on Physical Properties

In most cases, changes in parameters do not significantly alter the physical properties. For PBD simulations involving garment, the Particle Contact Offset parameter affects the thickness of the fabric; as its value increases, the fabric becomes progressively thicker. The Rest Offset parameter influences the distance between the dress and the ground upon landing, with an increase in this value resulting in a greater distance between the dress and the ground after it lands.

For PBD simulations involving fluid, the Velocity parameter affects the flow rate of the liquid; as its value increases, the liquid flows faster. The Cohesion parameter affects both the shape and flow rate of the liquid; at lower values, the liquid falls quickly and splashes out. As the value increases, the liquid flow slows down and splashing decreases, eventually leading to a smooth flow. The Particle Contact Offset parameter affects the form of the liquid as it falls; as its value increases, the liquid transitions from a continuous stream to a segmented, chunk-like flow.

In simulations involving deformable bodies, the Vertex Velocity Damping parameter affects the fall speed of objects such as hats; as the value increases, the fall speed decreases gradually. The Settling Threshold parameter also influences the fall speed of hats; increasing its value results in a slower fall speed, but once the value exceeds 1, the fall speed stabilizes. The Elasticity Damping parameter impacts the shape of the hat; as the value increases, the hat gradually collapses from a firm structure to a flat plane. The Young's Modulus parameter also affects the shape of the hat; at lower values (around 1e3), the hat collapses into a smaller height. As the value increases, the hat becomes firmer, and when the value reaches around 1e4, the hat initially stays firm and then gradually collapses. At a value of 15000, the hat remains completely firm.

For rigid body simulations, the Max Linear Velocity parameter affects the fall speed of rigid bodies such as hats; as the value increases, the fall speed decreases. When the value exceeds 50, the object practically stops falling.

In the context of flow simulations, the X-Component, Y-Component, and Z-Component parameters together determine the direction of the wind vector, while the Magnitude parameter determines the strength of the wind.

Table 7: Multi-physics parameters

| Type | Parameters | Function | Range |
|---|---|---|---|
| Garment | Particle Contact Offset | Distance at which particles start interacting | 0.03 - 0.12 |
| | Contact Offset | Distance at which collisions are detected | 0 - 16384 |
| | Rest Offset | Distance at which particles are in resting contact | 0 - 0.05 |
| | Solid Rest Offset | Distance for particle-solid interactions | Default |
| | Fluid Rest Offset | Distance for particle-fluid interactions | Default |
| | Solver Position Iteration Count | Number of iterations for solver to satisfy constraints | 6 - 255 |
| | Max Depenetration Velocity | Maximum speed at which particles are separated when overlapping | inf |
| | Max Neighborhood | Maximum number of neighboring particles for interactions | 36 - 512 |
| | Density | Mass per unit volume of the material | default |
| | Friction | Resistance to sliding motion | default |
| | Damping | Reduction of motion or oscillations | default |
| | Viscosity | Internal friction within the fluid material | default |
| | Cohesion | Attractive force between particles | default |
| | Surface Tension | Elastic tendency of the material's surface | default |
| | Drag | Resistance experienced when moving through fluid or air | default |
| | Lift | Force acting perpendicular to fluid flow around the material | default |
| Fluid | Particle Contact Offset | Distance at which particles start interacting | 0.17 - 0.3 |
| | Contact Offset | Distance at which collisions are detected | default |
| | Rest Offset | Distance at which particles are in resting contact | 0.03 - 0.2 |
| | Solid Rest Offset | Distance for particle-solid interactions | 0.1-0.2 |
| | Fluid Rest Offset | Distance for particle-fluid interactions | 0.1-0.15 |
| | Solver Position Iteration Count | Number of iterations for solver to satisfy constraints | 6 - 255 |
| | Max Depenetration Velocity | Maximum speed at which particles are separated when overlapping | inf |
| | Max Neighborhood | Maximum number of neighboring particles for interactions | 36 - 512 |
| | Density | Mass per unit volume of the material | 0 - 1e10 |
| | Friction | Resistance to sliding motion | 0 - 0.2 |
| | Damping | Reduction of motion or oscillations | 0 - 10 |
| | Viscosity | Internal friction within the fluid material | 1e3 - 1e6 |
| | Cohesion | Attractive force between particles | 0 - 100 |
| | Surface Tension | Elastic tendency of the material's surface | 0 - 100 |
| | Drag | Resistance experienced when moving through fluid or air | 0 - 78 |
| | Lift | Force acting perpendicular to fluid flow around the material | 0 - 1e10 |
| Deformable Body | Vertex Velocity Damping | Rate of reduction of vertex velocities | 0 - 10 |
| | Simulation Mesh Resolution | Granularity of the simulation mesh | 10 |
| | Solver Position Iterations | Number of iterations for solver to satisfy positional constraints | 8 - 255 |
| | Sleep Threshold | Velocity below which the body is considered to be at rest | 0 - 1e7 |
| | Settling Threshold | Velocity below which the body is considered to have settled | 0 - 1e7 |
| | Sleep Damping | Additional damping as the body approaches the sleep threshold | 0 - 1e7 |
| | Contact Offset | Distance at which collisions are detected | -inf |
| | Rest Offset | Distance at which particles are in resting contact | -inf |
| | Self Collision Filter Distance | Minimum distance to avoid self-collision | -inf |
| | Remeshing Resolution | Resolution for remeshing the input mesh | Default |
| | Target Triangle Count | Target resolution for quadric simplification | Default |
| | Max Depenetration Velocity | Maximum speed at which vertices can be separated when overlapping | inf |
| | Density | Mass per unit volume | Default |
| | Dynamic Friction | Resistance to sliding motion | 0 - 2048 |
| | Young's Modulus | Stiffness of the material | 1e3 - 1e10 |
| | Poisson's Ratio | Ratio of transverse to axial strain | 0 - 0.499 |
| | Elasticity Damping | Reduction of oscillations and vibrations | 0 - 0.05 |
| | Damping Scale | Adjusts the overall damping effect | 0 - 1.0 |
| Rigid Body | Max Linear Velocity | The rate of change of position of the rigid body. | 0-50 |
| | Max Angular Velocity | The rate of change of rotation of the rigid body. | 0-1e10 |
| | Collision Shape | Defines the shape used for collision detection. | default |
| | Contact Offset | Distance from the surface where collisions are detected. | -inf-inf |
| | Rest Offset | Effective contact distance from the surface. | -inf-inf |
| | Convex Hull | Approximation method for collision shape. | 0-64 |
| | SDF (Signed Distance Field) | Approximation method using signed distance field. | default |
| | Mass | Defines the mass of the rigid body. | 0 to Inf |
| | Density | Defines the density of the rigid body material. | 0 to 1000 |
| | Friction | Resistance to sliding motion. | 0 to 1 |
| | Restitution (Bounciness) | Degree of elasticity of collisions. | 0 to 1 |
| | Material Density | Density of the material applied to the rigid body. | 0 to 1000 |
| Flow | X-Component | Flow rate in the x-direction | -inf-inf |
| | Y-Component | Flow rate in the y-direction | -inf-inf |
| | Z-Component | Flow rate in the z-direction | -inf-inf |
| | Magnitude | Overall magnitude of the flow | 0-inf |

# E  Sim2Real

Transferring models trained in simulator to reality is challenging and become a critical issue for robotic research.However, most Sim2Real techniques are not yet fully automated and require careful human oversight. **In this work, we present three visual sim2real methods which are fully automated and self-supervised**. We mainly conduct experiment follow [57], learning dense visual representation for garment before and after our alignment method. The results are shown in Figure 6

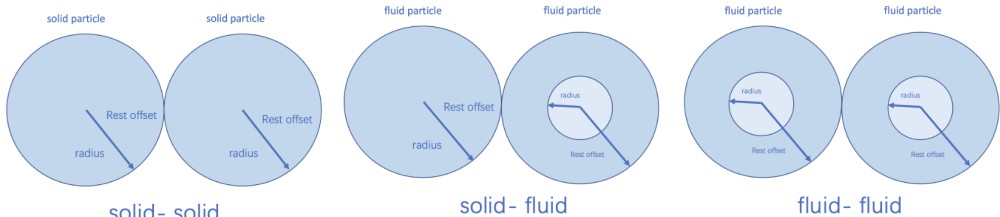

Figure 8: Different Types of Particle-Particle Interaction

### E.1  Sim-Real Vision Alignment

**Noisy Observation.**  Although previous do dedicated exploration on how to add noise to point cloud[26, 62], they need capture IR picture and do many calculation which is time-consuming. However, we have found that simply adding salt-and-pepper noise and Gaussian noise can already yield very good results. We directly add noise to depth picture and generate noised point cloud in the training data.

$$D_{\text{noised}}(x,y) = \begin{cases} D(x,y) + \mathcal{N}(0,\sigma^2) & \text{with probability } p_{\text{gaussian}} \\ D(x,y) + \text{salt} & \text{with probability } p_{\text{salt}} \\ D(x,y) - \text{pepper} & \text{with probability } p_{\text{pepper}} \\ D(x,y) & \text{otherwise} \end{cases}$$

where $D(x,y)$ represents the depth value at pixel $(x,y)$ in the original depth picture, $\mathcal{N}(0,\sigma^2)$ represents Gaussian noise with mean 0 and variance $\sigma^2$, and salt and pepper noise is added with probabilities $p_{\text{gaussian}}$, $p_{\text{salt}}$, and $p_{\text{pepper}}$ respectively.

For experiment, we add noise to the training data during the training process of dense visual correspondence[57]. As shown in figure6, before we do data argumentation for training data, the query results show many spots, indicating errors. This is due to discontinuities of dense representation caused by differences between the real-world point cloud and the simulator. After we add noise, the model become more robust to noise thus the representation become more smooth and accurate.

**Point Cloud Alignment.**  Dense object descriptors [14] that learn point- or pixel-level object representations are proposed by and for robotic manipulation. The key idea of these works[57, 54, 50]is to represent an object as a function f that maps a 3D coordinate x to a spatial descriptor $z = f(x)$ of that 3D coordinate:$f(x) : \mathbb{R}^3 \to \mathbb{R}^n$. $f$ may further be conditioned on point cloud $\mathbf{P} \in \mathbb{R}^{3 \times N}$ and usually parameterized by a neural network. However, $f$ are not always SE(3)-equivariant, which means to a rigid transform $(\mathbf{R},\mathbf{t}) \in SE(3)$, we can **NOT** guarantee that $f(\mathbf{x}|\mathbf{P}) \equiv f(\mathbf{R}\mathbf{x} + \mathbf{t}|\mathbf{R}\mathbf{P} + \mathbf{t})$. However, in real world experiment, as the height and angle of the camera may be different from that in the simulator, the distributions of point cloud collected in simulation and real world are different. This will lead to wrong query result especially for garment as it highly rely on thickness to detect folding relationships.

Although we can choose SO(3)-equivariant network[13], the training of it is hard and time-consuming. Thus, we propose a direct way to align the point cloud in simulation and the realworld. As all rigid transform $(\mathbf{R},\mathbf{t}) \in SE(3)$ can be represented by affine matrix, we directly use gradient descent to optimize the affine matrix so that we can align the distribution between realworld point cloud and simulation point cloud. We chose the chamfer distance as the loss function because it is both robust to the various deform and shape of the garment and effectively aligns the positions. Equation 1 show our optimization objective and 2 show our loss function.

$$\text{Transforms}(R,t) = \begin{pmatrix} a & b & c & x \\ d & e & f & y \\ g & h & i & z \\ 0 & 0 & 0 & 1 \end{pmatrix} \tag{1}$$

$$\text{Chamfer\_Loss}(A, B) = \frac{1}{|A|} \sum_{a \in A} \min_{b \in B} \|a - b\|^2 + \frac{1}{|B|} \sum_{b \in B} \min_{a \in A} \|a - b\|^2 \qquad (2)$$

As shown in figure 6, before we align the point cloud, the model predict wrong result even in the flat case. After alignment, the model successfully predict the correct result.

**Keypoint Embedding alignment.** As the model learn point level representation, a direct way is to align the representation of corresponding point between realworld garment and simulation. In this part, we first attach marker to garment on the skeleton points(shoulder, end of the sleeve and bottom corner etc.) Then, we do self-play which enable franka to randomly choose the pick-and-place point to create garment deformation status. Then we use SAM[24] to detect key point and align correspondence key point with the simulation result. After we get ground-truth key point pair between simulation and realworld,we employ InfoNCE[25] a widely-used loss function in one-positive-multi-negative-pair contrastive representation learning, to pull close the representation of corresponding points, while push away representation of them and other point representations. The loss function is shown in Equation 3

$$\mathcal{L}_{CD} = -log(\frac{exp(f_p \cdot f_{p'}/\tau)}{\sum_{i=1}^{m} exp(f_p \cdot f_{p'_i}/\tau)}) \qquad (3)$$

where $f_p$ is the skeleton point in realworld garment, $f_{p'}$ is the corresponding point in simulation point cloud and $f_{p'_i}$ is other point in simulation point cloud.

As shown in figure 6, after model finetune, the performance of the model on realworld garment improve significantly. This is mainly because model is more adapted to the distribution of real world point cloud.

# F  Real-World Benchmark

Benchmarking and performance evaluation in robotic manipulation encounter challenges owing to the diverse range of applications and tasks, prompting research groups to select representative tasks and objects that are frequently inadequately specified and inaccessible to others, thereby impeding the ability to compare experimental results and interpret performance quantitatively, particularly in real-world scenarios. To address this issue, the implementation of a real-world benchmark is crucial, as it can not only narrow sim2real Gap but also provide a platform for researchers to directly compare algorithm performance. Although previous work has introduced real-world benchmarks, such as YCB [5] and furniture benchmark [18], they primarily focus on rigid bodies, lacking benchmarks specifically designed for deformable objects. **In this study, we introduce the first real-world benchmark for deformable objects and garments, facilitating the widespread usage of a standardized set of objects and tasks to enable easy comparison of results among research groups worldwide.**

## F.1  Object and Data Set: Object Selection

**Principle**
We aimed to select objects that are frequently used in daily life, and we also reviewed the literature to consider objects that are frequently used in simulations and experiments. Several additional practical factors must be considered when formulating the proposed set of goals and tasks.

- *Variety*
  The objects included are small in number, but ensure a great richness. Judging from the category of items, we roughly include tops, pants, skirts, socks, gloves, dolls, etc. Considering size, generally speaking, clothes occupy a larger area, followed by dolls, and then small items such as socks and gloves. Considering deformability, large items of clothing are the softest, can be stacked into various shapes, and have the highest deformability, followed by small items, which can only undergo simple changes because they are relatively small, while dolls are elastic but lack deformability. Grasping and manipulation difficulty was also a criterion: for instance, toys are well approximated by simple geometric shapes and relatively easy for grasp synthesis and execution, while garments have higher shape complexity and are more challenging for grasp synthesis and

execution. In addition, since we are doing a benchmark about garments, we have to carefully consider their characteristics: they are diverse and highly deformable, but the same type of garment often only differs in texture or color, and is very similar in structure and key points. This allows us to use a few objects to represent a category of garments, thereby ensuring the variety of our benchmark.

- *Use*

  We included objects that are not only interesting for grasping but that also have a wide range of manipulation uses. Soft and highly deformable clothing is also suitable for many complex operations: such as hanging, folding, etc. The introduction of fluid allows us to simulate the interaction between some objects and fluids, such as washing and air-drying. In addition, we also included people, which allowed us to simulate the interaction between some objects and people, such as putting a scarf on someone. As mentioned above, these tasks are intended to span a wide range of difficulty, from relatively easy to very difficult.

- *Durability*

  We aimed for objects that can be useful in the long term, and, therefore, avoid objects that are fragile or perishable. In addition, to increase the longevity of the object set, we chose objects that are likely to remain in circulation and change relatively little in the near future.

- *Cost*

  We aimed to keep the cost of the object set as low as possible to broaden accessibility. We, therefore, selected standard consumer products, rather than, for instance, custom-fabricated objects, and tests. We buy all our clothes from Uniqlo and all our dolls from Disney.

After these considerations, the final objects were selected. You can see our object set in Table 8 and the corresponding suggested tasks in Table 9.

Table 9: **Suggested Manipulation Tasks for Different Garments.**

| Object Category | Suggested Tasks |
|---|---|
| Tops/Pants/Shorts/Vests/Skirts | a. Washing and Drying
b. Folding
c. Stacking and Grabbing of clothes |
| Dresses/Coats | a. Washing and Drying
b. Folding
c. Stacking and Grabbing of clothes
d. Hanging |
| Hats/Scarfs | a. Washing and Drying
b. Folding
c. Stacking and Grabbing of clothes
d. Hanging
e. Interacting with People: wearing corresponding objects on people |
| Deformable Objects | a. Grabbing and Placing |
| Small Items (Gloves/Stocks) | a. Grabbing and Placing
b. Washing and Drying |

### F.2 Object Scans

Our scanning process is roughly divided into four stages: model wearing clothes, scanner scanning to obtain raw data, post-processing, and manual annotation of key points. The whole process can be referred to Figure 9.

Table 8: **Real-World Objects and Properties**. PBD stands for Position Base Dynamics, while FEM stands for Finite Element Method.

| Number | Class | Object Name | Picture | Simulation Method | Branch | Number | Class | Object Name | Picture | Simulation Method | Branch |
|---|---|---|---|---|---|---|---|---|---|---|---|
| 1 | FEM Objects | Straw -berry Bear |  | FEM | Disney | 11 | Shorts | Blue Denim Shorts |  | PBD | Uniqlo |
| 2 | FEM Objects | Stitch |  | FEM | Disney | 12 | Shorts | White Shorts |  | PBD | Uniqlo |
| 3 | FEM Objects | Winnie Bear |  | FEM | Disney | 13 | Skirts | Blue Denim Shorts |  | PBD | Uniqlo |
| 4 | Coats | White Jacket |  | PBD | Uniqlo | 14 | Vests | White Cotton Vest |  | PBD | Uniqlo |
| 5 | Coats | Green Jacket |  | PBD | Uniqlo | 15 | Shorts | White Camisole Vest |  | PBD | Uniqlo |
| 6 | Coats | White Plush Jacket |  | PBD | Uniqlo | 16 | Dresses | Blue Dress |  | PBD | Uniqlo |
| 7 | Tops | White-long-sleeved T-shirt |  | PBD | Uniqlo | 17 | Shorts | White Blue Dress |  | PBD | Uniqlo |
| 8 | Tops | White-short-sleeved T-shirt |  | PBD | Uniqlo | 18 | FEM Objects | Blue Hat |  | FEM | Uniqlo |
| 9 | Pants | White Pants |  | PBD | Uniqlo | 19 | FEM Objects | Blue Socks |  | FEM | Uniqlo |
| 10 | Pants | Black Pants |  | PBD | Uniqlo | 20 | FEM Objects | Yellow Gloves |  | FEM | Uniqlo |

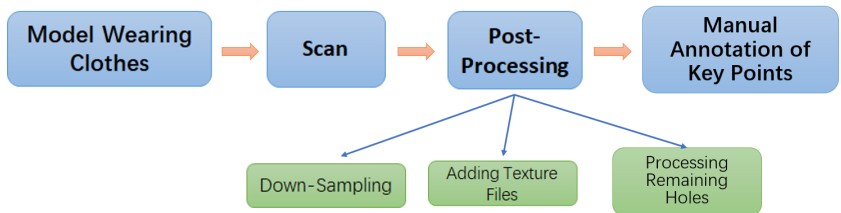

Figure 9: **Objects Scanning Process.**

- *Model wearing clothes*
  There are two main ways to scan clothes: scanning them flat and scanning them while the model is wearing them. After careful consideration and constant experimentation, we chose the latter. This is because we use a large number of points to simulate clothing, so the wrinkles formed when the clothes are laid flat will cause the points to be unevenly distributed, thus forming many "holes". While when worn on the model, the appearance of wrinkles will be reduced, thus try to avoid this situation as much as possible.

- *Scanning*
  We use a scanner to scan the object from multiple angles, and obtain a copy of the original point cloud data and grid data through the combination of the depth camera and the RGB camera. The general process can be referred to Figure 5.

- *Post-Processing*
  During the post-processing process, we mainly did three things: down-sampling, adding texture files, and processing remaining holes. The point set obtained by the initial scan has too many points and is difficult to support with ordinary computing power, so we performed down-sampling to generate a file that can retain the main features and have a moderate number of points. We use Meshlab for down-sampling. The original scale is about 100 million points and 300 million faces. After down-sampling, it can reach about 10,000 points and 30,000 faces. We use MTL files (Material Library File) to add material attribute information. MTL is a material library file used to describe the material information of objects. It is usually used in conjunction with an OBJ file to apply material properties such as texture and color to the OBJ model. In this step, we mainly implemented some visual textures, such as patterns, colors, etc. The physical texture is achieved through different simulation methods. In addition, there are still some "holes" in the processed point set, which we repaired manually.

- *Manual annotation of key points*
  Since we use a large number of points to simulate objects, it is necessary to mark some key points and edges to indicate important features. For example, for tops, we will mark the sleeves, neckline, hem, etc. These locations are often the key parts for clothing operations. For dolls, we will mark arms, legs and other parts that are easy to grasp. Figure 10 gives some examples for reference.

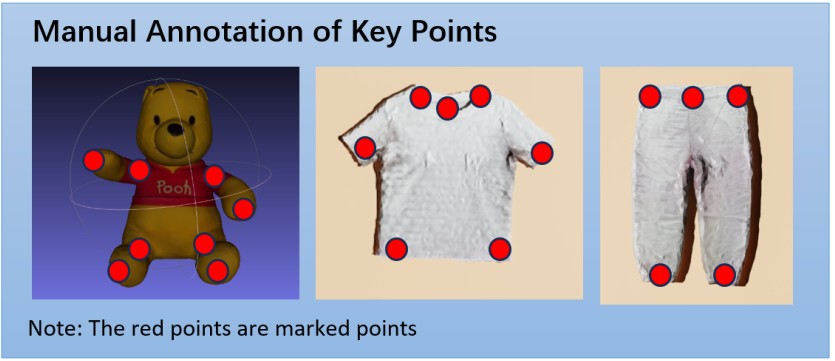

Figure 10: Manual annotation of key points: examples

### F.3 Protocol Benchmark Guidelines

We use the protocol and benchmark templates mentioned in [72]. To both provide more concrete samples of the types of task definitions that can be put forward as well as specific and useful benchmarks for actually quantifying performance, we have developed some example protocols: clothes-hanging protocol, scarf-wearing protocol.

- *Clothes-hanging protocol and benchmark*
  When dealing with flexible clothing, hanging clothing is always a popular task. The protocol uses the hanger, clothes suitable for hanging of our model set. The clothes are initially laid flat on the platform, and the robot is expected to grab the key points of the clothes, pick them up, and finally hang them on the hangers. The benchmark scores the performance of the robot by evaluating whether the hanging point is reasonable and the stability of the hanging clothes (whether they are easy to fall off). We applied this benchmark to Franka.

- *Scarf-wearing protocol*
  Dressing people with robots has always been a difficult subject. In this example, we chose the easier and more manageable task: wearing a scarf. This protocol uses the scarf from the model set, and a person from the Issac Simulator. The scarf is initially laid flat on the platform, and what the robot has to do is to pick it up and wrap it around the person's neck, and finally adjust it to a suitable state. The benchmark scores the performance of the robot by evaluating whether the final state of the scarf is stable (we can test it by adding wind to see if the scarf will blow off quickly), whether the remaining length of the scarf on both sides is similar, and whether the scarf fits the person's neck (rather than loosely packed). We applied this benchmark to Franka.

## G  Task

### G.1  Task category

**Garment-Garment.** This category focuses on fundamental garment manipulation, including tasks like folding and unfolding single garments, as well as interactions between multiple garments such as retrieving items from clothes piles. Tasks in this category include folding, unfolding (pick and place), and unfolding (fling).

**Garment-Fluid.** Tasks in this group concentrate on the interaction between garments and fluids as well as flow, where trajectory dynamics play a crucial role. This category of tasks includes washing clothes in a basin, rinsing clothes under running water, and drying clothes with a hairdryer. In this category, we specifically introduced the interaction between the robot manipulating objects and fluid flow, including both water flow and air flow.

**Garment-FEMObjects.** We mainly focus on the exploration of tasks involving deformable interactions, such as using a sponge to clean dirt off clothes or packing hats and tops together. Some simple tasks involving the manipulation of deformable objects are also included, such as using a dexterous hand or gripper to grab plush toys.

**Garment-Rigid.** Common interactions between clothing and rigid bodies, such as hanging clothes or putting them into a washing machine, require precise grasp point selection and trajectory planning.We also introduced articulated objects such as cabinet drawers and clips to perform garment-related tasks, such as taking clothes out of a wardrobe.

**Garment-Avatar.** Dressing tasks pose the greatest challenge, as they require understanding of human intention and safe collaboration with humans. Some representative tasks include putting a scarf on a person and placing a hat on their head. More advanced tasks involve dressing a person in a jacket or a T-shirt.

### G.2  Long-horizon task

**Organizing clothes.** This task comprises several stages, including retrieving tops or trousers from a clothes pile, unfolding them using fling, folding the clothes, and placing them in the wardrobe.

**Wash clothes.** This task involves several stages: retrieving a hat from the cabinet, washing the hat in the basin, using a hairdryer to dry the hat, and placing the hat on a hanger.

**Make up table.** The task of setting the table involves several steps: firstly, retrieving the tablecloth from the box, laying it flat, spreading it onto the table, smoothing it out, and finally adjusting its position. Note here we need the use of mobile robot like mobile franka.

**Dress up.** This task involves putting a scarf on someone, placing a hat on someone's head, and dressing someone in a T-shirt.

## H  MoveIt

We adopt MoveIt, an open-source state-of-the-art robotic manipulation framework, to provide support for real-world trajectories planning and obstacle avoidance. To record the trajectories generated by MoveIt and adapt visual models in the simulator to the trajectories could bridge the sim2real gap to some extent. In this section, we introduce a smooth, lightweight and responsive signal pipeline implementation to transfer real-world joint parameters to the simulator.

As a robotic manipulation platform, MoveIt is built on top of ROS(Robot Operating System) and integrates with various ROS components. MoveIt provides a series of comprehensive manipulation interfaces, including collision-free motion planning, kinematics computation, collision detection, etc. Moreover, real-world data collected from sensors like cameras and lidars can be fed into MoveIt, allowing for dynamic obstacle avoidance. Once the trajectories are generated by MoveIt, we publish the computed joint states through ROS, which transfers the trajectory to the Franka controller, FrankaPy. FrankaPy is a modular control stack that provides a customizable and accessible interface to the Franka robot. Utilizing MoveIt and FrankaPy, this pipeline enables Franka to devise a collision-free path and guide the gripper to the target posisition using vision detectors, while publishing the joint parameters to the ROS server. The simulator then subscribes the joint states and moves the Franka model accordingly.

## I  Teleoperation

Teleoperation serves as a direct method to acquire human demonstrations for model training. To accurately and smoothly track human hand motions has been proved advantageous in related works recently. However, the proliferated fine-grained tracking requirements, along with sparse and diverse dexterous hand models and environment settings, have posed a challenge towards teleoperation systems. Compared to controller-based models, we utilize the vision-based motion detection module, Leap Motion, to efficiently record human hand poses and then retarget hand poses to the dexterous robot hand. More Formally, our teleopertaion systems can be described as below:

(i) **Leap Motion hand pose detection module**, which predicts the wrist position, the hand spatial direction and finger poses from the infrared camera stream.

(ii) **retargeting module**, which converts the wrist position and finger poses recorded by Leap Motion to the arm end effector position and dexterous hand parameters.

(iii) **motion generating module**, which produces accurate, responsive and high-frequency signals for the robot model that in the simulator and in the real world simultaneously.

In our work, we implement teleoperation for Universal Robot mounted with Shadow Hand and Franka. One can control the posture of the robot by adjusting attitude and position of the hand over the detector. In particular, the open state of the gripper of Franka can be controlled by the opening and closing of the thumb and index finger.

### I.1  Leap Motion Detection Module

The Leap Motion Controller is a small USB device that can be placed on the desktop. Utilizing two 640x240-pixel near-infrared cameras, it captures a roughly hemishperical area in the distance of approximately 60 cm, typically at 120Hz. The internal algorithms then translate the received raw spatial data to 27 distinct hand elements, which includes the palm normal vector, the hand direction, the wrist position and 24 finger joint positions. The detailed hand elements are shown in the figure 12:

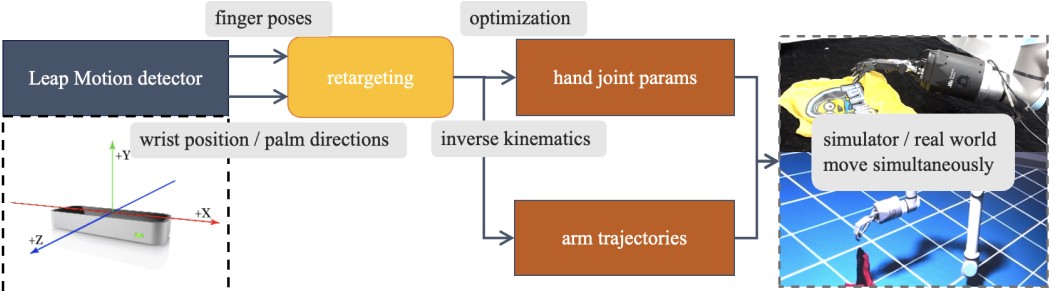

Figure 11: **Details of Our Teleoperation System.**

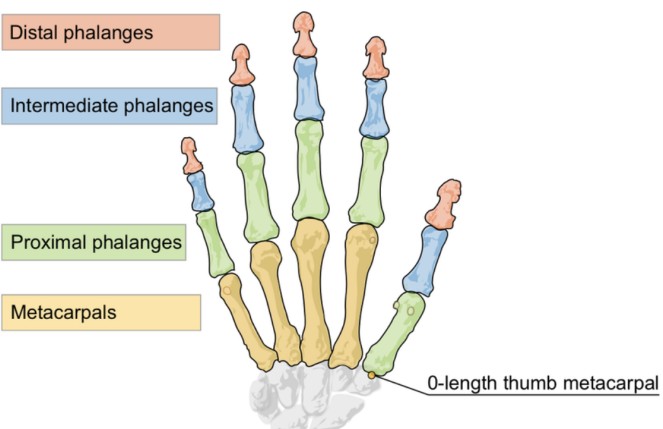

Figure 12: **The Hand Model Used in Leap Motion**, each knuckle joints' spatial position is computed by retargeting algorithm and broadcasted though ROS Message.

## I.2 Hand Pose Retargeting

The procedure of hand pose retargeting is two-fold: first, we map knuckle positions to hand joint parameters; second, we compute a trajectory to smoothly move the robot arm to the recorded wrist position and direction. The finger knuckle positions captured by Leap Motion cannot be directly fed into robot models due to the discrepancy between the dexterous hand joint angular parameters and the knuckle positions. The mapping algorithm that converts knuckle positions to joint parameters is often formulated as an optimization problem, which can be described as

$$\min_{q_t} \sum_{i=0}^{N} \left\| \alpha v_t^i - f_i(q_t) \right\|^2 + \beta \left\| q_t - q_{t-1} \right\|$$

where $q_t$ represents joint parameters of the dexterous robot hand at time step $t$. $f_i$ is the $i$-th forward kinematic function which takes the joint angles as input and computes the knuckle positions. $\alpha$ is a scaling factor to alleviate the size discrepancy between different human operators and the robot hand model. Additionally, we observe adjacent $N$ frames and add hyperparameter $\beta$ to improve temporal smoothness and consistency.

To compute the arm trajectory, we adopt a slightly tweaked inverse kinematics approach, which is popular to determine the joint parameters in the trajectory, given the URDF file (Unified Robotics Description Format) of the robot and the desired configuration. URDF is a file format that describes the physical properties and 3D model of the robot, including joints, motors, articulation configurations, etc. In empirical experiments, we find that even slight hand movement or vibration can trigger a significant and prolonged changes in arm posture. To address this problem, we implement a rate limit on target changing in neighboring frames.

It is notable that this workflow is applicable to a vast range of grasp-based robots. Particularly, to control the open state of the gripper of Franka, we measure the distance of the thumb and index finger and establish a distance threshold in implementation.

### I.3 Motion Generation

We integrate ROS for communication. ROS is a open-source framework that provides a series of library which are designed for multi-robots scenarios. In our implementation, after the generation of hand pose and wrist position, the computed joint parameters are transferred through the ROS to the simulator and the non-virtual robot simultaneously.

## J    Limitation

While this work adopts state-of-the-art simulation for different garments, there still exists the gap between the dynamics and kinematics of garments in simulation and the real world.

## K    Broader Impact

This work paves solid way to future home-assistant robots in diverse garment tasks for industry. We haven't observed negative potential impacts.

