# OpenReview forum: "GarmentLab: A Unified Simulation and Benchmark for Garment Manipulation"
_NeurIPS.cc/2024/Conference — NeurIPS 2024 poster_

### Official Review · Reviewer_6kGg · 2024-07-10

**Soundness:** 3
**Presentation:** 3
**Contribution:** 3
**Rating:** 6
**Confidence:** 3

**Summary:**

The authors present GarmentLab, a content-rich benchmark and realistic simulation designed for deformable object and garment manipulation. Multiple tasks such as interactions between garments, deformable objects, rigid bodies, fluids, and human body are explored. A first real-world deformable benchmark along with several sim2real methods is also presented in this paper. The integration of ROS and MoveIt is also supported.

**Strengths:**

- The proposed framework is multi-functional supporting simulation of various interactions and coupling for garment manipulation scenarios. In addition to simulation, several sim-2-real algorithms such as Teleoperation, Visual Sim-Real Alignment are provided. The integration with ROS and MoveIt also help the framework to be a off-the-shelf tool for the community to use.
- Extensive evaluations and experiments are presented.
- The paper is well-written and organized.

**Weaknesses:**

The main contributions of this paper lie in the engineering and benchmarking (or dataset) aspects. Although I acknowledge the substantial contribution to the community, the novelty of this work remains a concern.

**Questions:**

See weaknesses.

**Limitations:**

Yes.

---

> ### Author Rebuttal · Authors · 2024-08-07
>
> Thank you for your feedback! We have addressed your comments below.
>
>
>
> ## Novelty of Our Paper
>
> We are grateful to your review on our **contribution**: "the substantial contribution to the community".
>
> As for the **novelty** of our paper,
> as our paper proposes the environment and benchmark for robotic manipulation,
> the novelties mostly lie in the comparisons with other environments and benchmarks.
>
> First, garment manipulation is a novel research direction.
> While previous studies mostly provide environments for rigid objects, or relatively simple deformable objects (e.g., cables, rings and fabrics), our work first comprehensively provides the simulation and benchmark for the novel garment manipulation direction.
>
> Second, the diversity of objects, interactions and tasks is our novelty.
> While previous environments mostly only support the simulation of a certain type of object or interaction,
> our work not only supports simulation for rigid and articulated objects,
> but also supports different simulation methods for different deformable objects.
> For example, Position-Based Dynamics (PBD) [1] for shirts, Finite Element Method (FEM) [2] for hats and gloves, and even the simulation of fluid and flow.
> For these diverse object simulation methods, we further support the interaction coupling among them.
> The simulation of these diverse objects and their interactions provides a rich environment for various tasks,
> which has not been achieved in previous benchmarks.
>
> Additionally,
> while previous benchmarks mostly focus on evaluation in simulation,
> our work provides a systematic sim-to-real pipeline and the real-world benchmark, paving way to the real-world deployment, comparison and benchmarks.
>
>
> [1] Bender, et al. "Position-Based Simulation Methods in Computer Graphics." Eurographics (tutorials). 2015.
>
> [2] Ciarlet, et al. The finite element method for elliptic problems. Society for Industrial and Applied Mathematics, 2002.
>
>
> We hope that we have clarified our novelty well. We are happy to take further questions from you!

---

> > ### Comment · Reviewer_6kGg · 2024-08-11
> > **Reply to rebuttal**
> >
> > Thanks for the reply to my concerns. I don't have further questions and will maintain my score.

---

### Official Review · Reviewer_1QRi · 2024-07-11

**Soundness:** 3
**Presentation:** 2
**Contribution:** 3
**Rating:** 6
**Confidence:** 4

**Summary:**

This paper proposes a content-rich benchmark and realistic simulation for deformation object and garment manipulation. The GarmentLab consists of the GarmentLab Engine, Assets, and Benchmark, supporting various research topics.

**Strengths:**

* The GarmentLab provides a realistic and rich environment for research topics related to garments.
* The engine supports different interactions between garments and other objects.
* The author further benchmarks various garment manipulation tasks based on this environment.
* Different sim2real methods are integrated into the GarmentLab.

**Weaknesses:**

Please refer to the questions.

**Questions:**

1. Will all the 3 components, i.e. the Engine, Assets and Benchmark, be publicly available?
2. As for the physics engine, is it possible to simulate the interactions among different layers of garments? And will different layers of garments really affect each other instead of generating dynamics layer by layer? For example, a human wearing multiple layers of garments doing customized actions.
3. What about the interactions between garments and fluid? Will the garment dynamics also be affected through the interactions with fluid? Some simulation engines do not support the real interactions between garments and fluid. They may generate the garment dynamics first, fix the dynamic sequence, and then simulate the fluid. Can the proposed engine solve this issue?
4. Is it possible to generate customised dynamics using the proposed physics engine? For example, using customised garments and interacting with other deformable objects.

**Limitations:**

Please refer to the questions.

---

> ### Author Rebuttal · Authors · 2024-08-07
>
> Thanks for your valuable comments! We have addressed them below, and hope to hear back from you
> if you have further questions!
>
> ## Code Release
>
> These mentioned components with the guidelines for installation and usage have been already made publicly available. Please refer to the **Global Response** for more details. Welcome to have a try!
>
> ## Interactions among Multiple Garment Layers
> GarmentLab supports simulating the interactions among different layers of garments, and these layers really affect each other.
> We use position-based dynamics (PBD) [1] to simulate garments with multiple layers as a whole to achieve this feature.
> As demonstrated in Figure 2 in the **Attached Rebuttal PDF**,
> the garment consists of multiple layers,
> and when the robot is trying to grasp and lift the front layer (from the left sub-figure to the right sub-figure),
> other layers will have the effects resulted from the robot's operation on the grasped front layer.
>
>
> ## Interactions between Garments and Fluid
> GarmentLab supports simulating interactions between garments and fluid, and generating the dynamic sequences at the same time.
> The engine simulates garments and fluid using the same simulation method, position-based dynamics (PBD) [1], and thus supports this feature.
> As demonstrated in Figure 3 in the **Attached Rebuttal PDF**,
> in the first four sub-figures,
> we adjust the weight of the garment particles to make the garment respectively floats on the water, suspends in the water, and sinks to the bottom.
> This interaction requires importing both garment particles and fluid particles into the same particle system, and then generating dynamic sequences simultaneously.
> In the first four images, we adjust the weight of the clothing particles to make the clothes float on the water, suspend in the water, and sink to the bottom.
> Besides, we can also simulate clothing and fluid in different particle systems separately. As shown in the last sub-figure, the garment and fluid do not have interactions in this case.
> Additionally, we can adjust simulation parameters, such as particle contact offset and fluid rest offset, to simulate different interaction effects between garments and fluid. You can refer to Section D in **Appendix** for more details.
>
>
> ## Customized Dynamics
> GarmentLab supports customized dynamics. We can customize the dynamics in two ways. The **first** way to use customized meshes of garments or deformable objects. For example, in the left of Figure 4 in the **Attached Rebuttal PDF**, we import
> a garment mesh from our simulation assets and a deformable object mesh from our real-world benchmark. These two objects use different simulation methods: the garment uses Position-Based Dynamics (PBD), while the deformable object uses the Finite Element Method (FEM)[2].
> We can simulate the interactions between these two objects. This demonstrates that we can combine simulation objects and scanned objects from the real world.
> The **second** way is to adjust the parameters of the physical simulation to change the effect of object simulation. You can refer to Figure 3 in the **Main Paper** to see the effects of physics and Section D in **Appendix** for the physical parameters that can be adjusted.
> In summary, our GarmentLab is very flexible. It supports customized meshes of deformable objects and garments, and also allows for adjusting object parameters to modify the physical simulation effects.
>
>
> [1] Bender, et al. Position-Based Simulation Methods in Computer Graphics. Eurographics (tutorials). 2015.
>
> [2] Ciarlet, et al. The finite element method for elliptic problems. Society for Industrial and Applied Mathematics, 2002.

---

> ### Author Response · Authors · 2024-08-12
> **Look forward to hearing from you**
>
> Thank you again for your constructive feedback! As the discussion period is coming to an end, we want to know whether our responses have addressed your concerns, and if you have further questions. Thank you!

---

### Official Review · Reviewer_E1Nw · 2024-07-12

**Soundness:** 3
**Presentation:** 3
**Contribution:** 3
**Rating:** 7
**Confidence:** 3

**Summary:**

This paper proposes GarmentLab, a content-rich benchmark and realistic simulation specifically designed for deformable objects and garment operations. It encompasses a diverse range of garment types, robotic systems, and manipulators.

**Strengths:**

1. The GarmentLab Environment offers a realistic and comprehensive setting for garment manipulation. Additionally, the GarmentLab Benchmark covers a wide range of garment manipulation tasks and serves as the first real-world benchmark for these activities.
2. Although this job is primarily focused on engineering, it makes a certain contribution to the development of the garment manipulation community.

**Weaknesses:**

1. The paper mentions a characteristic of  GarmentLab is efficiency (Line 59), however, there are no quantifiable comparison with other enviroment in the speed and GPU memory during training/testing.
2. The modal of the GarmentLab environment is limited to visual (RGB-D), and other modalities such as touch and hearing can also be helpful when operating garment objects. Supporting this will help further development.

**Questions:**

See weaknesses.

**Limitations:**

The paper mentions that the limitation is the sim2real gap, perhaps with the development of simulation engines, this problem will be alleviated.

---

> ### Author Rebuttal · Authors · 2024-08-07
>
> Thanks for your valuable comments! We have addressed them below, and hope to hear back from you
> if you have further questions!
>
> # Efficiency of the Benchmark
> To the best of our knowledge, GarmentLab is the first to support all of ray tracing, robot control simulation, and diverse deformable object simulation methods in a whole environment.
> Therefore, it is a little bit difficult to directly compare the speed and memory of our environment with those of other environments in a fair setting.
> For example, SoftGym [1] and PlasticineLab [2] do not support real robot control (they only support pickers or manipulators as robots), which requires solving inverse kinematics and controlling the robot.
> FluidLab [3] does not support real time tracing, which is important for realistic visual effects.
> Furthermore, the above environments only support a single type of deformable object simulation method.
>
> While our environment supports all of the above features,
> we also take the speed and efficiency of the environment into account.
> GarmentLab is GPU-based, placing all simulations of deformable objects and real-time tracing on the GPU, while handling robot-related inverse kinematics on the CPU. This improves the simulator's speed and avoids the bottleneck of CPU-GPU data transfer.
> Moreover, to improve data collection performance, we support importing multiple robot-object interaction scenarios in the same simulation environment for parallel data collection.
> Please refer to 2:37 to 2:41 in the **Supplementary Video** for the demonstrations of simultaneous multi-scenario data collection.
>
> We have designed two settings to compare the speed of GarmentLab (with and without the multi-scenario data collection method respectively denoted as GarmentLab-Multiple and GarmentLab-Single) and SoftGym.
> First, we evaluate the speed of the robot reaching the target garment and then grasping the garment.
> Second, we evaluate the speed of the robot directly grasping the garment in the near front without other movement.
> The following tables respectively show the speed comparisons in these two settings,
> demonstrating that GarmentLab, with the designed multi-scenario data collection method, is comparable and a little more efficient than SoftGym in terms of simulation for data collection.
>
> | Environment | Data Collection Speed |
> |---------------------|-----------------------|
> | GarmentLab-Single            | 30 samples/hour        |
> | GarmentLab-Multiple             | 160 samples/hour       |
> | SoftGym             | 140 samples/hour     |
>
>
>
> | Environment | Data Collection Speed |
> |---------------------|-----------------------|
> | GarmentLab-Single            | 180 samples/hour      |
> | GarmentLab-Multiple             | 1200 samples/hour     |
> | SoftGym             | 900 samples/hour     |
>
> Thanks again for this valuable comment! This is really significant for a robotic manipulation environment.
> We will add the above descriptions, comparisons and tables in the final version of our paper.
>
>
> [1] Lin, et al. Softgym: Benchmarking deep reinforcement learning for deformable object manipulation. CoRL, 2021.
>
> [2] Zhou, et al. Fluidlab: A differentiable environment for benchmarking complex fluid manipulation. ICLR, 2023.
>
> [3] Huang, et al. PlasticineLab: A Soft-Body Manipulation Benchmark with Differentiable Physics, ICLR, 2021.
>
>
>
>
> # More Input Modalities
> Thanks for this valuable comment!
> The current engine supports tactile detection for rigid-rigid interaction.
> As demonstrated in Figure 4 (Right) in the "Attached Rebuttal PDF",
> we replaced the Franka gripper with the Trossen's X300 gripper and installed tactile sensors on its inner side,
> and can detect the tactile feedback including normal and friction points and forces between the robot manipulator and the rigid object using SDF collision.
> With this feature, we are able to design interesting tasks like using a rigid stick to nudge and adjust the posture of garments. The clothing provides force feedback to the stick, which is then perceived by the tactile sensors attached to the manipulator.
> Furthermore, we are actively extending and developing this feature into directly sensing the tactile feedback of garments,
> which could largely support more interesting and fine-grained manipulation tasks,
> and we are willing to present such features in our future work, to continuously make contributions to the community.
>
> For other modalities, the engine also supports IMU, Lidar, Radar, Ultrasonic and Generic Range sensors, while these sensors are rarely used in garment manipulation tasks.
> If some tasks require such modalities, we are super willing to extend such sensory signals to our proposed envrionment.

---

> > ### Comment · Reviewer_E1Nw · 2024-08-09
> > **Official Comment by Reviewer E1Nw**
> >
> > Thanks for the well-written rebuttal. I keep my score and support the acceptance of this paper.
> >
> > Looking forward to the release of the code and hoping that you can continue to maintain it.

---

### Official Review · Reviewer_wEM9 · 2024-07-13

**Soundness:** 3
**Presentation:** 3
**Contribution:** 3
**Rating:** 6
**Confidence:** 4

**Summary:**

This paper introduces GarmentLab, a new set of garment manipulation environments, assets, annotations and tasks based on IssacSim. The differences compared to existing benchmarks is the support of a broader range of tasks (such as tasks involving interaction with other objects such as hanger, fluid, and human avatar), more robot types, more assets, and other features such as motion planning, teleoperation, and sim2real transfer. A real-world garment manipulation benchmark is also proposed with the aim for evaluating real-world performances. Several existing algorithms are tested on the proposed tasks, and their performances are analyzed.

**Strengths:**

- The overall system looks very comprehensive and cover most of the needs for garment manipulation, including a broad range of tasks, assets, and robot control methods.
- The sim2real analysis and support for real-world data collection are nice features.
- Overall the whole system looks impressive and can be a good contribution to the deformable object manipulation community.

**Weaknesses:**

I mainly have the following questions:
- Regarding the real-world benchmarks: I appreciate the authors provide the 3d scans of the real objects and provide semantic annotations. However, I think the proposed real-world benchmark still lacks details such that every lab can easily replicate the set up in the real world. For example, it would be ideal to include purchase links for the chose objects. The protocols presented in F.3 seems to be just describing the task in a very high-level way, without details such as how the robot / objects should exactly be initialized, e.g., their relative position, and how to exactly measure the success / performance. Without such details it would not be possible for other labs to perfectly replicate this real-world setups and to form a fair comparison for different algorithms. I would encourage the authors to provide more details on such protocols to ensure that this can be well replicated in any lab, otherwise this proposed real-world benchmark would seem infeasible.
- For some readers not familiar with the context, it is hard to understand the right part of figure 6. How can one interpret this figure to understand that the performance becomes better after applying the sim2real method such as point cloud align? E.g., what does the color scheme mean, and why is the color scheme after better than before? Also what is the figure in the dashed bounding box supposed to show? I would suggest to revise this figure to make these more clear.
- Since this is a benchmarking and dataset paper, it is a little bit hard to access its quality and potential contribution to the community without actually using/seeing the code. For example, numerous robotics benchmarking has been proposed in the past, but only a few of them will be widely adopted by the community, where factor such as easy-to-use APIs, intuitive UI design, informative debugging tools, are all critical for the survival of the benchmarking. As the code of this paper is not released, it's hard to access the above qualities. I would encourage the author to release the code as soon as possible, and take into consideration the above factors, especially given the proposed system seems complicated and involve many different components.
- An error in table 1: PyBullet actually support garments. See Antonova et al., Dynamic Environments with Deformable Objects, NeurIPS Datasets and Benchmarks Track, 2021. This paper is also highly relevant and should probably be discussed in related work.
- There are some typos in the paper, e.g., line 6 in the abstract: "which exhibit offer limited diversity" -> "which offer limited diversity". Please have a thorough check over the whole paper for typos.

**Questions:**

Please see the weakness sections.

**Limitations:**

Yes.

---

> ### Author Rebuttal · Authors · 2024-08-07
>
> Thanks for your valuable comments. We have addressed them below, and hope to hear back from you
> if you have further questions!
>
> # Setup of Real-World Evaluation
>
> Thanks for the appreciation of our proposed real-world assets,
> and pointing out that the real-world evaluation setup is not detailed enough.
> We have attached detailed descriptions (task, setup, hardware requirement, execution procedure, evaluation methods, etc.) about the real-world implementation protocols
> in the released document of GarmentLab. Specifically, the Protocol sub-section in the Real-World Benchmark section of the released document.
>
> Thanks again for this valuable comment!
> We hope that the real-world benchmark can really help the community to evaluate the real-world performance of garment manipulation methods.
>
> # Clarifications on Figure 6
>
> For the **color scheme** of Figure 6, it denotes the per-point correspondence between garments.
> According to [1], when the learned representations of two points in two garments are similar, the point colors would be similar.
> Such per-point correspondence is a good way to both (1) evaluate and visualize the learned representation of garments in an intuitive way,
> and (2) further guide the manipulation of novel garments within few-shot demonstrations in previous garments, by selecting the manipulation points on the novel garment, which are the most similar to those in the demonstrations garments.
> More details of such representation can be found in [1].
> When adding sim-to-real methods (Noise Injection for Domain Randomization, Corresponding Keypoint Representation Alignment, Point Cloud Alignment between Simulation Garments and Real-World Scans),
> the learned representations, reflected by the color scheme, can be more continuous and smooth across different point on the different garments.
>
> We have revised Figure 6 (as Figure 1 in the "Attached Rebuttal PDF"),
> showing the procedures of three sim-to-real methods on the left of the green block, and there corresponding effects on the learned representations of garments (making the learned representations more continuous and smooth, with better correspondence across different garments) on the right of the green block.
>
> Thanks again for this valuable comment! We are sparing no effort to make our paper more clear.
>
> ## Details of Code and Usage
>
> As described in the "Global Response", we have made GarmentLab public, with the documents of installation and usage. Welcome to have a try!
>
> We are very grateful for your tips, and we have taken the details you mentioned into account when designing the benchmark.
> For **UI design**, we adopted the Isaac Sim UI interface. As a mature, industrial-grade simulator, Isaac Sim's UI is easy to operate, as verified by practical tests, making it much more user-friendly than most other simulators' basic interfaces.
> Moreover, it supports moving with the mouse during operation, which is something most simulators [2, 3] cannot do.
> Additionally,
> the functions that can be achieved in the UI interface can easily be extended to **code APIs**,
> and we have provided detailed instructions on how to use the APIs in the code and documentation,
> with many example tasks to control the robot and achieve the goal.
> For **debugging tools**, Isaac Sim and Omniverse already have a very comprehensive warning and error reporting system.
> On this basis, we have added numerous assertions at key points in our code to ensure that user input meets the requirements and to ensure that users correctly understand how to use the API.
> It is worth noting that, as Isaac Sim is still under active development,
> it sometimes encounters segmentation faults without detailed descriptions.
> We have included more detailed error or warning statements in our high-level code.
> This way, users can quickly locate the problem when an error occurs.
>
>
> # Descriptions on PyBullet
>
> Thank you for pointing out that [4] based on PyBullet also supports the simulation of garment manipulation tasks.
> Compared to Pybullet based methods, our proposed GarmentLab has the following advantages:
>
> 1. GarmentLab supports the coupling of garments simulated by different simulation method (PBD and FEB), and the interactions between garments with different materials (e.g., hat, gloves and shirts).
> On the contrary, previous works with garments simulated in PyBullet could only support one kind of simulation, without the simulation of multi-material interactions.
> 2. GarmentLab includes GPU acceleration for simulation in the large scale, which supports collecting large scale data and training in an efficient manner.
> 3. Equipped with high fidelity GPU based PhysX engine, GarmentLab is capable of supporting multi-sensor RTX rendering at an industrial scale.
> 4. While [4] only demonstrated 4 representative tasks on only a few categories of garments, we present and benchmark 11 garment categories with much more diverse tasks.
>
> We will include the above descriptions in the final version of the paper.
>
>
> # Typos
> Thanks a lot for pointing out this. We have thoroughly checked our paper, and corrected them.
>
>
>
> [1] Wu, et al. UniGarmentManip: A Unified Framework for Category-Level GarmentManipulation via Dense Visual Correspondence. CVPR 2024.
>
> [2] Lin, et al. Softgym: Benchmarking deep reinforcement learning for deformable object manipulation. CoRL, 2021.
>
> [3] Zhou, et al. Fluidlab: A differentiable environment for benchmarking complex fluid manipulation. ICLR, 2023.
>
> [4] Antonova, et al. Dynamic Environments with Deformable Objects, NeurIPS Datasets and Benchmarks Track, 2021.

---

> > ### Comment · Reviewer_wEM9 · 2024-08-11
> > **Official Comment**
> >
> > I want to thank the authors for the detailed and well-written rebuttal. I don't have further questions, and I remain positive about this paper.

---

### Author Rebuttal · Authors · 2024-08-07

We extend our gratitude to reviewers for their careful reading, meticulous feedback and valuable insights!

We are glad that reviewers unanimously agree that our work:

(1) proposes a comprehensive and realistic environment for garment manipulation (wEM9: "very comprehensive, cover most of the needs", E1Nw: "comprehensive, realistic, content-rich; a wide range of tasks", 1QRi:"realistic and rich", 6kGg: "multi-functional, various interactions and coupling");

(2) designs well for sim-to-real transfer (wEM9: "sim2real analysis and support for real-world data collection are nice features", E1Nw: "first real-world benchmark", 1QRi: "different sim2real methods are integrated", 6kGg: "several sim-2-real methods are provided");

(3) makes good contributions to the community (wEM9: "a good contribution to the deformable object manipulation community", E1NW: "makes a certain contribution to the development of the garment manipulation community", 6kGg: "substantial contribution to the community").

In this **Global Response TEXT**, we clarify some common concerns raised by reviewers. Thanks again for valuable comments! Any further questions are welcomed!

## Code Release (Reviewer wEM9 and 1QRi)

We have made GarmentLab publicly available, and sent the corresponding anonymous link to the Area Chair.
The open-sourced components include：
+ Instructions for installation of the environment and benchmark based on the Engine
+ All simulation assets (scenes, robots, garments, scenes and other objects)
+ Real-World Benchmark: scans and purchase links of real-world garments, setup of real-world manipulation
+ Code for loading robots, garments and scenes
+ Code for robot control
+ Code for more than 10 benchmark tasks
+ Code for teleoperation
+ Code for customizing the physics of the simulated garments
+ Code for data collection and learning baselines (point-level correspondence and affordance for manipulation)
+ Documents (User Manual) of how use the environment, including detailed descriptions of different components and corresponding demos

Welcome to have a try on GarmentLab! We are making every effort to create a user-friendly, high-quality and powerful benchmark.

Furthermore, we are also actively developing the environment to support more robots, tasks, sensors and modalities. We hope to do our best to contribute high-quality, easy-to-use garment manipulation environment and benchmark to the community.

For the remaining questions, we will respond to each reviewer individually. We appreciate the valuable and insightful comments from reviewers! If you have any further questions or suggestions, please feel free to contact us. We are more than happy to discuss with you!

---

### Decision · Program_Chairs · 2024-09-25

**Decision:**

Accept (poster)

**Comment:**

This submission introduces GarmentLab, a comprehensive benchmark and realistic simulation environment for garment manipulation, featuring diverse tasks, multiple simulation methods, and sim-to-real algorithms, designed to advance research in deformable object manipulation for home-assistant robots. The system is based on NVIDIA IssacSim using FEM and PBD methods. Codes are submitted and planned to be released. All the four reviews vote to accept the submission. The AC sees the submission as a clear acceptance.